# Sequential phosphorylation of NDEL1 by the DYRK2-GSK3β complex is critical for neuronal morphogenesis

Youngsik Woo[1], Soo Jeong Kim[1], Bo Kyoung Suh[1], Yongdo Kwak[1†],
Hyun-Jin Jung[2], Truong Thi My Nhung[1], Dong Jin Mun[1], Ji-Ho Hong[1‡],
Su-Jin Noh[1], Seunghyun Kim[1], Ahryoung Lee[1], Seung Tae Baek[1],
Minh Dang Nguyen[3,4,5,6], Youngshik Choe[2], Sang Ki Park[1*]

[1]Department of Life Sciences, Pohang University of Science and Technology, Pohang, Republic of Korea; [2]Korea Brain Research Institute, Daegu, Republic of Korea; [3]Hotchkiss Brain Institute, Cumming School of Medicine, University of Calgary, Calgary, Canada; [4]Department of Clinical Neurosciences, Cumming School of Medicine, University of Calgary, Calgary, Canada; [5]Department of Cell Biology and Anatomy, Cumming School of Medicine, University of Calgary, Calgary, Canada; [6]Department of Biochemistry and Molecular Biology, Cumming School of Medicine, University of Calgary, Calgary, Canada

*For correspondence:
skpark@postech.ac.kr

Present address: †SK Biopharmaceuticals Ltd, Republic of Korea; ‡LG Chem Ltd, Republic of Korea

Competing interests: The authors declare that no competing interests exist.

**Abstract** Neuronal morphogenesis requires multiple regulatory pathways to appropriately determine axonal and dendritic structures, thereby to enable the functional neural connectivity. Yet, however, the precise mechanisms and components that regulate neuronal morphogenesis are still largely unknown. Here, we newly identified the sequential phosphorylation of NDEL1 critical for neuronal morphogenesis through the human kinome screening and phospho-proteomics analysis of NDEL1 from mouse brain lysate. DYRK2 phosphorylates NDEL1 S336 to prime the phosphorylation of NDEL1 S332 by GSK3β. TARA, an interaction partner of NDEL1, scaffolds DYRK2 and GSK3β to form a tripartite complex and enhances NDEL1 S336/S332 phosphorylation. This dual phosphorylation increases the filamentous actin dynamics. Ultimately, the phosphorylation enhances both axonal and dendritic outgrowth and promotes their arborization. Together, our findings suggest the NDEL1 phosphorylation at S336/S332 by the TARA-DYRK2-GSK3β complex as a novel regulatory mechanism underlying neuronal morphogenesis.

## Introduction

Establishment of neuronal morphology is a key process during neurodevelopment. The neuronal morphogenesis process involves the extension and the branching of axons and dendrites in order to allow each neuron to determine functional connections with other neurons. Indeed, perturbations of this process can cause severe deficits in brain functions in various neurodevelopmental disorders such as autism spectrum disorder, attention deficit hyperactive disorder, and schizophrenia (*Birnbaum and Weinberger, 2017*; *Forrest et al., 2018*; *Schubert et al., 2015*). Although its complex regulatory pathways composed of various players are still largely unknown, the orchestrated remodeling of the cytoskeleton is a crucial step for neuronal morphogenesis (*Coles and Bradke, 2015*; *Dent and Gertler, 2003*; *Rodriguez et al., 2003*).

Nuclear distribution element-like 1 (NDEL1) plays multifaceted roles in neurodevelopmental processes (*Chansard et al., 2011*). NDEL1 expression in the nervous system begins at early embryonic stage and is maintained throughout the adulthood (*Pei et al., 2014*; *Sasaki et al., 2000*). Deficiency in *Ndel1* results in embryonic lethality (*Sasaki et al., 2005*) and postmortem studies and human

genetic studies have implicated NDEL1 in several neuropsychiatric diseases such as schizophrenia (*Bradshaw and Hayashi, 2017*; *Burdick et al., 2008*; *Gadelha et al., 2016*; *Lipska et al., 2006*; *Nicodemus et al., 2010*), both emphasizing the importance of NDEL1 functions in brain development. In the developing brain, NDEL1 regulates neuronal precursor proliferation and differentiation (*Liang et al., 2007*; *Stehman et al., 2007*; *Ye et al., 2017*), neuronal migration (*Okamoto et al., 2015*; *Sasaki et al., 2005*; *Shu et al., 2004*; *Takitoh et al., 2012*; *Youn et al., 2009*), and neuronal maturation (*Hayashi et al., 2010*; *Jiang et al., 2016*; *Kamiya et al., 2006*; *Kuijpers et al., 2016*; *Saito et al., 2017*; *Shim et al., 2008*; *Youn et al., 2009*). These functions are supposed to be regulated by multiple post-translational modifications (PTMs), but the detailed mechanism underlying them is yet fully understood.

NDEL1 directly binds to Trio-associated repeat on actin (TARA, also known as TRIOBP isoform 1) (*Hong et al., 2016*), a short isoform of Trio-binding protein (TRIOBP) generated by alternative splicing (*Riazuddin et al., 2006*; *Seipel et al., 2001*). TARA associates with filamentous actin (F-actin) and has functions in cell mitosis and cell migration (*Hong et al., 2016*; *Seipel et al., 2001*; *Zhu et al., 2012*). Although its abnormal aggregation has also been observed in the postmortem brains of patients with schizophrenia (*Bradshaw et al., 2014*; *Bradshaw et al., 2017*), the role of the TARA in neurodevelopment remains largely unknown. Furthermore, the molecular mechanisms underlying functions of NDEL1-TARA complex have yet to be unraveled.

Here, we introduced the large-scale human kinome library screening and the unbiased LC-MS/MS analysis of NDEL1 in order to systematically search regulatory mechanisms for its functions in brain development. We identified the novel sequential phosphorylation at S336 and S332 by DYRK2 and GSK3β and its function in neuronal morphogenesis, particularly in axon/dendrite outgrowth and neuronal arborization, through modulation of F-actin dynamics. We propose a new signaling mechanism that TARA scaffolds DYRK2 and GSK3β and recruits them to NDEL1, thereby inducing sequential phosphorylation of NDEL1 S336/S332 that is crucial for establishing the neuronal morphology. Taking together, our results provide a new biological insight to understand underlying mechanism for neuronal morphogenesis thereby for relevant neurodevelopmental disorders.

## Results

### DYRK2 and GSK3β induce sequential phosphorylation of NDEL1 at S336 and S332

In order to search regulatory mechanisms toward NDEL1 functions, we screened the human kinome library (Center for Cancer Systems Biology (Dana Farber Cancer Institute)-Broad Human Kinase ORF collection) for kinases responsible for NDEL1 phosphorylation (*Johannessen et al., 2010*; *Yang et al., 2011*). NDEL1 phosphorylation was determined by the band shift assay that has been shown to be effective in detecting phosphorylation of NDEL1 (*Niethammer et al., 2000*; *Yan et al., 2003*). Among the 218 serine/threonine kinases tested, we identified dual specificity tyrosine-phosphorylation-regulated kinase 2 (DYRK2) and homeodomain-interacting protein kinase 4 (HIPK4) as the candidates (*Figure 1—figure supplement 1*). DYRK2 and HIPK4 belong to the DYRK family and share an evolutionarily conserved DYRK homology (DH)-box (*Arai et al., 2007*; *Aranda et al., 2011*; *Becker et al., 1998*). Moreover, DYRK family kinases often act as priming kinases for GSK3 on various targets (*Cole et al., 2006*; *Nishi and Lin, 2005*; *Taira et al., 2012*; *Woods et al., 2001*). Indeed, DYRK2 induced significant phosphorylation of NDEL1 while co-expression of DYRK2 and GSK3β$^{S9A}$, a constitutively active form of GSK3β, induced at least two phosphorylation bands (*Figure 1A*). In silico analysis predicted NDEL1 S336 and S332 residues as the target sites by DYRK2 and GSK3β, respectively. When we tested each phospho-deficient mutant, S332A (NDEL1$^{S332A}$) and S336A (NDEL1$^{S336A}$), NDEL1$^{S332A}$ retained a single phosphorylation band by DYRK2, but lost GSK3β effect, while either NDEL1$^{S336A}$ or S332A/S336A double mutant (NDEL1$^{S332/336A}$) showed no detectable phosphorylation band (*Figure 1B*, *Figure 1—figure supplement 2A*). It verified a sequential phosphorylation process, first at S336 by DYRK2 followed by at S332 by GSK3β.

We then assessed the PTMs of endogenous NDEL1 proteins isolated from postnatal day 7 (P7) developing mouse brain by LC-MS/MS analysis to see whether the S332 and 336 can be phosphorylated *in vivo* (*Figure 1C*, *Figure 1—figure supplement 3*). Interestingly, among multiple PTMs

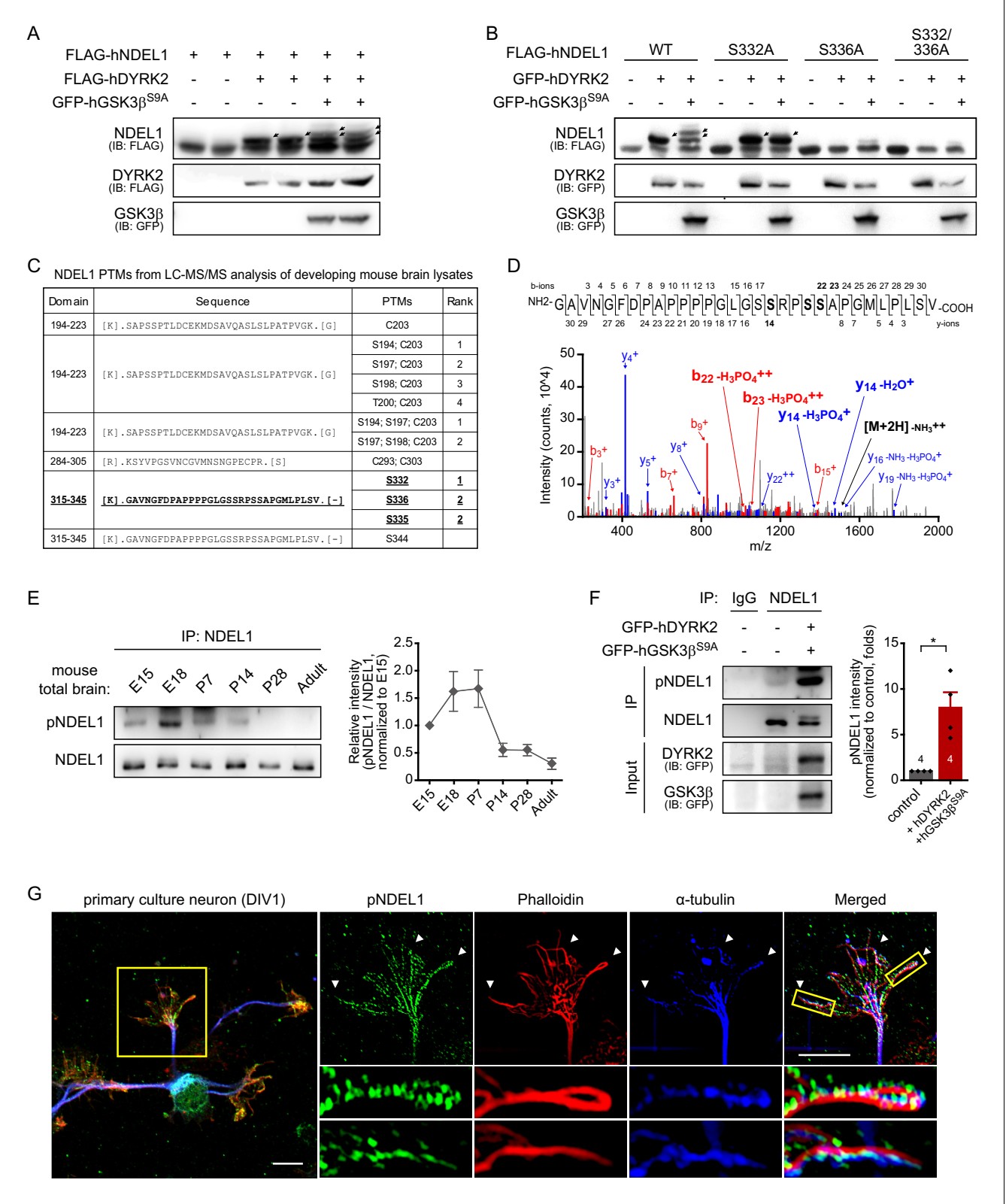

**Figure 1.** DYRK2 and GSK3β induce sequential phosphorylation of NDEL1 at S336 and S332. (**A**) Identification of responsible kinases for NDEL1 phosphorylation. DYRK2, one of the positive candidates from human kinome screening for NDEL1 phosphorylation, and GSK3β[S9A] sequentially phosphorylated NDEL1. (**B**) Identification of NDEL1 S336 and S332 as target sites of DYRK2 and GSK3β. DYRK2 increased NDEL1 phosphorylation at S336 and GSK3β[S9A] additionally induced phosphorylation at S332. (**C**) List of NDEL1 PTMs identified from LC-MS/MS analysis of developing mouse

*Figure 1 continued on next page*

*Figure 1 continued*

brain lysates. Peptide containing S336 and S332 phosphorylation is indicated by bold and underlined letters. (**D**) MS/MS spectrum for the phosphorylated fragments of NDEL1 peptide including S332 and S336 residues. The sequence of the peptide (aa 315–345) and all detected fragment ions are shown above. The b- and y-ions annotated in the spectrum include the sizes of $y_{14}$ and $b_{22}$ ions indicative of phosphorylation at S332 or S336. (**E**) Phosphorylation levels of endogenous NDEL1 S336/S332 in the developing mouse brain. Amount of lysates subjected to IP was normalized by NDEL1 protein levels. N = 7 for E15, P7, P14, P28, and adult brain lysates and N = 6 for E18. All results are presented as means ± SEM. (**F**) Increased endogenous NDEL1 phosphorylation by DYRK2 and GSK3β. Transfected HEK293 cell lysates were IPed with pan-NDEL1 antibody followed by western blot with anti-pNDEL1 antibody. Over-expression of DYRK2 and GSK3β$^{S9A}$ increased the endogenous NDEL1 S336/S332 phosphorylation. The number of samples is shown at the bottom of the bar of the graph. All results are presented as means ± SEM. *p<0.05 from Student's t-test. (**G**) Endogenous NDEL1 S336/S332 phosphorylation detected at the growth cone of the primary cultured mouse hippocampal neuron. Anti-pNDEL1 antibody signal was enriched at the growth cone and colocalized with both phalloidin and α-tubulin (indicated by arrowheads). Magnified confocal images show the strong overlap between pNDEL1, phalloidin, and α-tubulin. The scale bar represents 10 μm. See also *Figure 1—figure supplements 1*, *2* and *3* and *Figure 1—source data 1* and *2*.

The online version of this article includes the following source data and figure supplement(s) for figure 1:

**Source data 1.** Source data for quantitation of endogenous pNDEL1 in developing mouse brain lysates.

**Source data 2.** Source data for quantitation of endogenous pNDEL1 with kinases over-expression.

**Figure supplement 1.** Figure of kinome library screening results for NDEL1 phosphorylation CCSB-Broad.

**Figure supplement 2.** Figure related to *Figure 1*.

**Figure supplement 3.** Table for the list of NDEL1 PTMs identified from LC-MS/MS analysis of developing mouse brain lysates.

identified, we observed masses indicating phosphorylation at NDEL1 S332 or S336 (*Figure 1D*), suggesting the potential roles of NDEL1 phosphorylation in neurodevelopmental processes.

To monitor the phosphorylation state of NDEL1 S336 and S332, we raised an NDEL1 S336/S332 phosphorylation-specific antibody (anti-pNDEL1). As detected by anti-pNDEL1 antibody, the phosphorylation was decreased in NDEL1$^{S332A}$ and virtually absent in NDEL1$^{S336A}$ (*Figure 1—figure supplement 2B*). Anti-pNDEL1 antibody also detected bands corresponding to endogenous NDEL1 proteins immunoprecipitated (IPed) from lysates of developing mouse brain (*Figure 1—figure supplement 2C*). In the brain lysates from various developmental stages, pNDEL1 signal peaked at embryonic day 18 (E18) and P7 (*Figure 1E*), the stage where residual neuronal migration and intense neurite outgrowth and maturation occur. Additionally, the increment of the endogenous NDEL1 phosphorylation by DYRK2-GSK3β$^{S9A}$ expression was detected by anti-pNDEL1 antibody (*Figure 1F*).

The pNDEL1 signal in mouse brain slices was significantly diminished upon NDEL1 knockdown by *in utero* electroporation of an shRNA construct, further validating the immunostaining specificity of the antibody (*Figure 1—figure supplement 2D*). In cultured primary hippocampal neurons, the significant pNDEL1 signal was detected in the growth cone, a region of dynamic crosstalk between actin and microtubules (*Figure 1G*). This crosstalk is required for correct neurite extension and branching (*Coles and Bradke, 2015*; *Pacheco and Gallo, 2016*; *Rodriguez et al., 2003*). Moreover, the pNDEL1 signal overlapped with that of both phalloidin, a marker for F-actin, and α-tubulin. Notably, phosphorylated NDEL1 was prominently present in filopodia-like structures where both F-actin and microtubules are colocalized (*Figure 1G*, arrowheads).

## Phosphorylation of NDEL1 S336/S332 regulates neuronal morphogenesis

The phosphorylated NDEL1 was colocalized with both actin and microtubules at the growth cone, thus we next examined the roles for NDEL1 S336/S332 phosphorylation in the neuronal development. In cultured hippocampal neurons, NDEL1 knockdown significantly reduced both the total neurite length and the longest neurite length by about 31% and 33%, respectively (*Figure 2A–C*). Co-expression of an shRNA-resistant form of NDEL1$^{WT}$, but not NDEL1$^{S332/336A}$, effectively reversed these phenotypes providing specificity to the NDEL1 phosphorylation. These NDEL1 phosphorylation-specific effects were found for both axonal and dendritic neurites (*Figure 2—figure supplement 1A–B*). In addition, co-expression of DYRK2 and GSK3β$^{S9A}$ with NDEL1$^{WT}$, but not with NDEL1$^{S332/336A}$, significantly increased the total neurite length and the longest neurite length by about 22% and 19%, respectively (*Figure 2D–F*, *Figure 2—figure supplement 1C–D*). Furthermore, to avoid potential complications caused by an overexpression of NDEL1, we applied CRISPR/Cas9

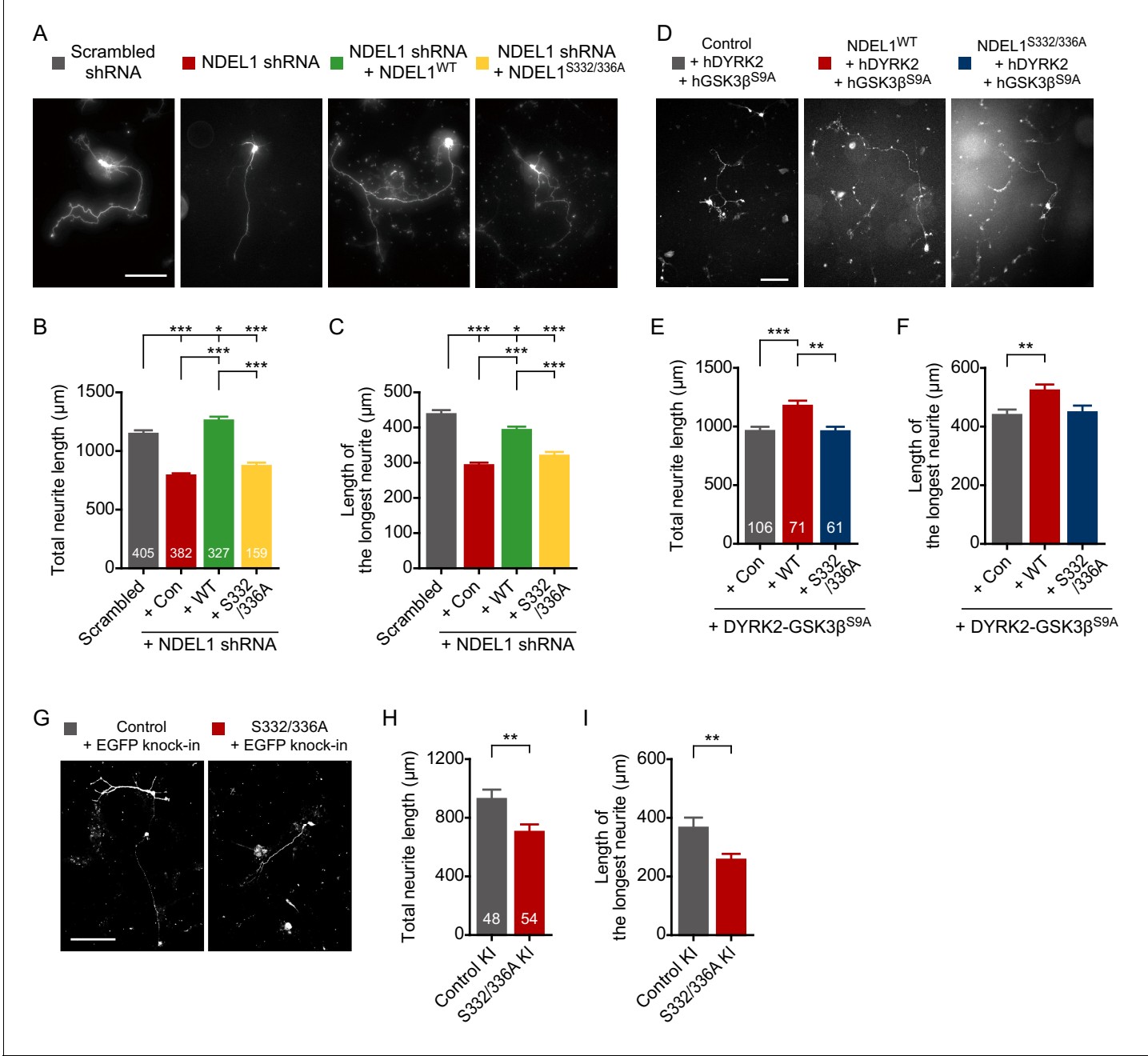

**Figure 2.** Phosphorylation of NDEL1 S336/S332 increases neurite outgrowth of primary cultured hippocampal neurons. (**A–C**) Suppression of NDEL1 S336/S332 phosphorylation inhibited neurite outgrowth of hippocampal neurons. Each DNA construct was transfected at 9 hr after neuronal culture and neurites of transfected neurons were analyzed at DIV 3. All of NDEL1 over-expressing constructs here contain an shRNA-resistant mutation. (**A**) Representative images of transfected neurons. The total neurite length (**B**) and the longest neurite length (**C**) were measured by using ImageJ software. (**D–F**) NDEL1 S336/S332 phosphorylation induced by DYRK2-GSK3β co-expression is enough to increase neurite outgrowth of hippocampal neurons. Each DNA construct was transfected at DIV 2 and neurites of transfected neurons were analyzed at DIV 5. (**D**) Representative images of transfected neurons. The total neurite length (**E**) and the longest neurite length (**F**) were measured by using ImageJ software. (**G–I**) KI of phospho-deficient mutations resulted in neurite outgrowth defects. DNA constructs for KI were transfected at DIV 1 and neurites of transfected neurons were analyzed at DIV 4. (**G**) Representative images of transfected neurons. The total neurite length (**H**) and the longest neurite length (**I**) were measured by using ImageJ software. Each sample number is shown at the bottom of the bar of the graph. Scale bars represent 100 μm. All results are presented as means ± SEM. *p<0.05, **p<0.01, and ***p<0.001 from one-way ANOVA for (**B**), (**C**), (**E**), and (**F**) and Student's t-test for (**H**) and (**I**). All experiments were independently repeated at least three times. See also *Figure 2—figure supplements 1* and *2* and *Figure 2—source data 1*, *2* and *3*.
The online version of this article includes the following source data and figure supplement(s) for figure 2:

*Figure 2 continued on next page*

*Figure 2 continued*

**Source data 1.** Source data for axon/dendrite outgrowth of NDEL1 knockdown and rescue groups.
**Source data 2.** Source data for axon/dendrite outgrowth of NDEL1 and kinases over-expression groups.
**Source data 3.** Source data for axon/dendrite outgrowth assay of NDEL1 S332/336A KI.
**Figure supplement 1.** Figure for additional data on axon/dendrite outgrowth.
**Figure supplement 2.** Schematic diagram of NDEL1 S332/336A KI strategy.

based knock-in (KI) approach targeting mouse *Ndel1* exon nine in order to generate a phospho-deficient NDEL1 S332/336A mutation with insertion of EGFP as the KI marker (*Figure 2—figure supplement 2*). NDEL1 S332/336A KI resulted in the diminished neurite lengths of the primary cultured neurons (*Figure 2G–I*, *Figure 2—figure supplement 1E–F*). Taken together, these results show that NDEL1 phosphorylation at S336 and S332 up-regulates axon/dendrite outgrowth.

Suppression of NDEL1 expression in mouse brain decreases the number of dendritic branches (*Saito et al., 2017*). In *Drosophila*, the loss of NudE, a homolog of both NDE1 and NDEL1, results in abnormal dendritic arborization (*Arthur et al., 2015*). Thus, we next examined the roles of NDEL1 phosphorylation in the arborization of dendrites in the developing mouse brain. NDEL1 S336/S332 phosphorylation was prominently detected in neurons of the cortical layers II/III in P14 mouse brain (*Figure 3—figure supplement 1A*). Thus, we analyzed the dendritic structure of the pyramidal neurons in layers II/III at P14. NDEL1 knockdown decreased the number of intersections in Sholl analysis and the total length of dendrites (*Figure 3A–C*, *Figure 3—video 1*), in addition to the abnormal positioning of cortical neurons as previously reported (*Figure 3A*, arrowheads). Co-expression of an shRNA-resistant form of NDEL1[WT], but not NDEL1[S332/336A], reversed this defect in dendritic arborization. The total number of dendritic branches decreased upon suppression of NDEL1 phosphorylation with no significant change in the longest dendritic branch length (*Figure 3D*, *Figure 3—figure supplement 1B*). The effect was similar at both apical and basal dendrites (*Figure 3—figure supplement 1C–F*). In addition, suppression of NDEL1 phosphorylation resulted in decreased numbers of secondary and tertiary dendrites (*Figure 3E*). Conversely, over-expression of NDEL1[WT] and responsible kinases increased the total length and the branching number of dendrites with more intersections in Sholl analysis (*Figure 3F–J*, *Figure 3—figure supplement 1G–K*, and *Figure 3—video 2*). We also utilized human ubiquitin C (UBC) promoter to drive the expression of transgene at lower level and observed comparable effect of the phospho-deficient mutants (*Figure 2—figure supplement 1G–K* and *Figure 3—figure supplement 1L–N*). NDE1, a paralog of NDEL1 lacking C-terminal region harboring S336 and S332 residues, failed to effectively rescue neurite defects caused by NDEL1 KD (*Figure 2—figure supplement 1L–O*). Although *in utero* electroporation is intrinsically cell-type nonspecific and thus we cannot fully exclude the contribution of neuron-nonautonomous effects from neural or glial progenitors, these results collectively indicate that NDEL1 phosphorylation at S336 and S332 plays critical roles for neuronal arborization.

## TARA recruits DYRK2 and GSK3β to induce sequential phosphorylation of NDEL1 S336/S332

The interaction between NDEL1 and TARA has been identified previously (*Hong et al., 2016*). Interestingly, TARA co-expression introduced at least two additional forms of NDEL1 bands distinct in size (*Figure 4A*). Moreover, deletion of the NDEL1-interacting domain, hTARA[Δ413-499] (*Hong et al., 2016*) or mTARA[Δ401-487], effectively blocked the band shift (*Figure 4—figure supplement 1A–B*). To test whether the band mobility shifts are caused by PTMs, we treated protein phosphatase *in vitro* and it significantly diminished the mobility shifts of NDEL1 bands (*Figure 4A*). In addition, TARA enhanced the signals detected by phosphoserine-specific antibody (anti-PhosphoSerine) (*Figure 4—figure supplement 1C*). These results indicate that the NDEL1-TARA interaction promotes NDEL1 phosphorylation.

To characterize the TARA-dependent NDEL1 phosphorylation, we utilized LC-MS/MS analysis of FLAG-NDEL1 proteins upon co-expression of NDEL1 and TARA in HEK293 cells (*Figure 4—figure supplement 2*). Among the multiple PTMs detected, a phospho-peptide containing S336 residue was present (*Figure 4B*). The phosphorylations of both S336 and S332 in a single peptide were not recovered, in part due to technical limitations of LC-MS/MS. When we tested each single alanine

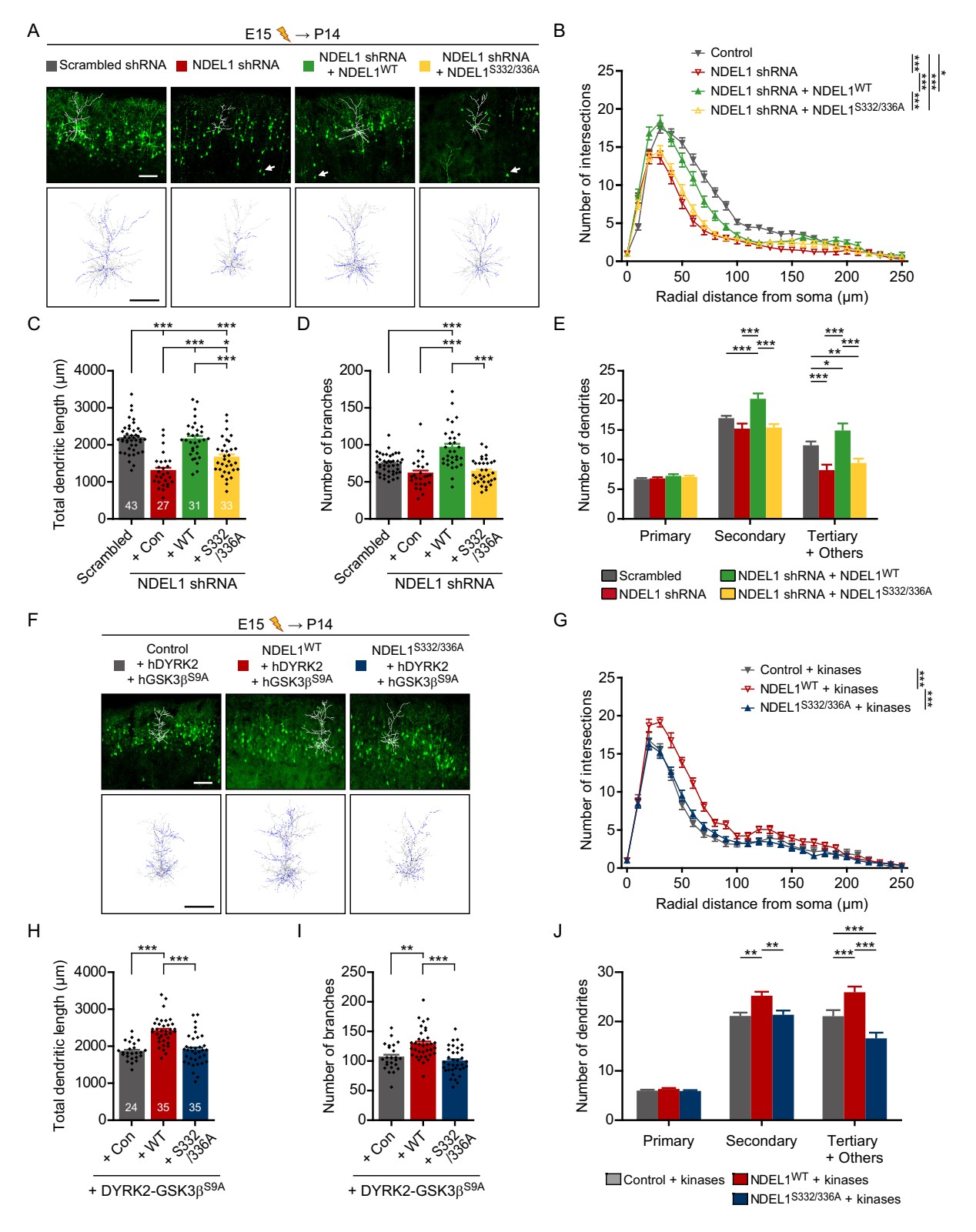

**Figure 3.** Phosphorylation of NDEL1 S336/S332 regulates dendritic arborization of cortical pyramidal neurons. (**A–E**) Suppression of NDEL1 S336/S332 phosphorylation disrupted dendritic arborization of layer II/III pyramidal neurons. All constructs were electroporated *in utero* to E15 mouse brain and then P14 brain was subjected for analysis. All of NDEL1 over-expressing constructs here contain an shRNA-resistant mutation. (**A**) Representative images of the brain slices with the tracked neuron (above) and the overlapped dendritic structures of five independent neurons (bottom). Sholl analysis

*Figure 3 continued on next page*

Figure 3 continued

plots (B), the total length of dendrites (C), the total number of branches (D), and the number of primary/secondary/tertiary dendrites (E) were analyzed by using Simple neurite tracer plug-in of ImageJ software. White arrowheads in (A) indicate neurons with migration defect. (F–J) NDEL1 S336/S332 phosphorylation induced by DYRK2-GSK3β kinases increased dendritic arborization of layer II/III pyramidal neurons. All constructs were electroporated *in utero* to E15 mouse brain and P14 brain was subjected for analysis. (F) Representative images of the brain slices with the tracked neuron (above) and the overlapped dendritic structures of five independent neurons (bottom). Sholl analysis plots (G), the total length of dendrites (H), the total number of branches (I), and the number of primary/secondary/tertiary dendrites (J) were analyzed by using Simple neurite tracer plug-in of ImageJ software. Each n number is shown at the bottom of the bar of the graph. Scale bars represent 100 µm. All results are presented as means ± SEM. *p<0.05, **p<0.01, and ***p<0.001 from one-way ANOVA for (C), (D), (H), and (I) and two-way ANOVA for (B), (E), (G), and (J). All brain samples for each group were collected from offspring of at least three independent *in utero* electroporation surgeries. See also *Figure 3—figure supplement 1*, *Figure 3—videos 1* and *2*, and *Figure 3—source data 1* and *2*.

The online version of this article includes the following video, source data, and figure supplement(s) for figure 3:

**Source data 1.** Source data for dendritic arborization of NDEL1 knockdown and rescue groups.
**Source data 2.** Source data for dendritic arborization of NDEL1 and kinases over-expression groups.
**Figure supplement 1.** Figure for additional data on dendritic arborization.
**Figure 3—video 1.** Video for 3D reconstructions of the dendritic structures of representative neurons, related to *Figure 3A*.
https://elifesciences.org/articles/50850#fig3video1
**Figure 3—video 2.** Video for 3D reconstructions of the dendritic structures of representative neurons, related to *Figure 3F*.
https://elifesciences.org/articles/50850#fig3video2

---

mutant, NDEL1$^{S332A}$ and NDEL1$^{S336A}$ showed decreased TARA-dependent phosphorylation (*Figure 4C*, *Figure 4—figure supplement 1D*). Particularly, NDEL1$^{S332A}$ showed only a single phosphorylation band, while NDEL1$^{S336A}$ showed a significant reduction in both phosphorylation bands.

To assess how TARA increases NDEL1 phosphorylation by DYRK2-GSK3β kinases, we tested protein-protein interactions among TARA and the kinases. Endogenous TARA was co-IPed with both DYRK2 and GSK3β (*Figure 4D–E*). Furthermore, over-expression of TARA enhanced the interaction between DYRK2 and GSK3β, suggesting that TARA acts as a scaffold (*Figure 4F*). Immunocytochemistry analysis confirmed their colocalization, further supporting their functional association (*Figure 4G*).

In cultured neurons, the co-expression of NDEL1$^{WT}$, but not NDEL1$^{S332/336A}$, with TARA significantly increased both the total neurite length and the longest neurite length (*Figure 5A–C*, *Figure 5—figure supplement 1A–B*). On the other hand, co-expression of NDEL1$^{WT}$ and hTARA$^{Δ413-499}$, which did not induce notable NDEL1 phosphorylation, did not increase the total neurite length and the longest neurite length (*Figure 5D–F*, *Figure 5—figure supplement 1C–D*). Consistently, in P14 brain slices, only neurons expressing NDEL1$^{WT}$ with TARA had a greater total dendrite length and more branches dependently to the phosphorylation (*Figure 5G–K*, *Figure 5—figure supplement 1E–I*, and *Figure 5—video 1*).

Thus, these results indicate that TARA recruits DYRK2 and GSK3β to induce NDEL1 S336/S332 phosphorylation, thereby enhancing neuronal morphogenesis.

## Phosphorylation of NDEL1 S336/S332 enhances F-actin dynamics

We then looked for the underlying mechanism by which NDEL1 S336/S332 phosphorylation increases axon/dendrite outgrowth and neuronal arborization. Since we observed anti-pNDEL1 antibody signal overlapping with both F-actin and microtubule at the growth cone (*Figure 1G*), we tested if NDEL1 S336/S332 phosphorylation affects cytoskeletal dynamics.

TARA directly binds to and stabilizes the F-actin structure and it recruits NDEL1 toward F-actin (*Hong et al., 2016*; *Seipel et al., 2001*). When we isolated F-actin from soluble G-actin via F-actin fractionation protocol, NDEL1 S336/S332 phosphorylation induced by TARA was detected in both insoluble F-actin fraction and soluble G-actin fraction, verifying that it can associate with F-actin structure (*Figure 6A*) comparable to its colocalization at growth cone (*Figure 1G*). Furthermore, we employed fluorescence recovery after photobleaching (FRAP) in combination with transfection of RFP-LifeAct to measure F-actin dynamics in the growth cone-like structure of differentiating SH-SY5Y cells (*Belin et al., 2014*; *Riedl et al., 2008*). NDEL1 knockdown suppressed the fluorescence recovery without changing the half-maximum recovery time (*Figure 6B–F*, *Figure 6—figure supplement 1A–D*, and *Figure 6—video 1*). Co-expression of an shRNA-resistant form of NDEL1$^{WT}$, but not

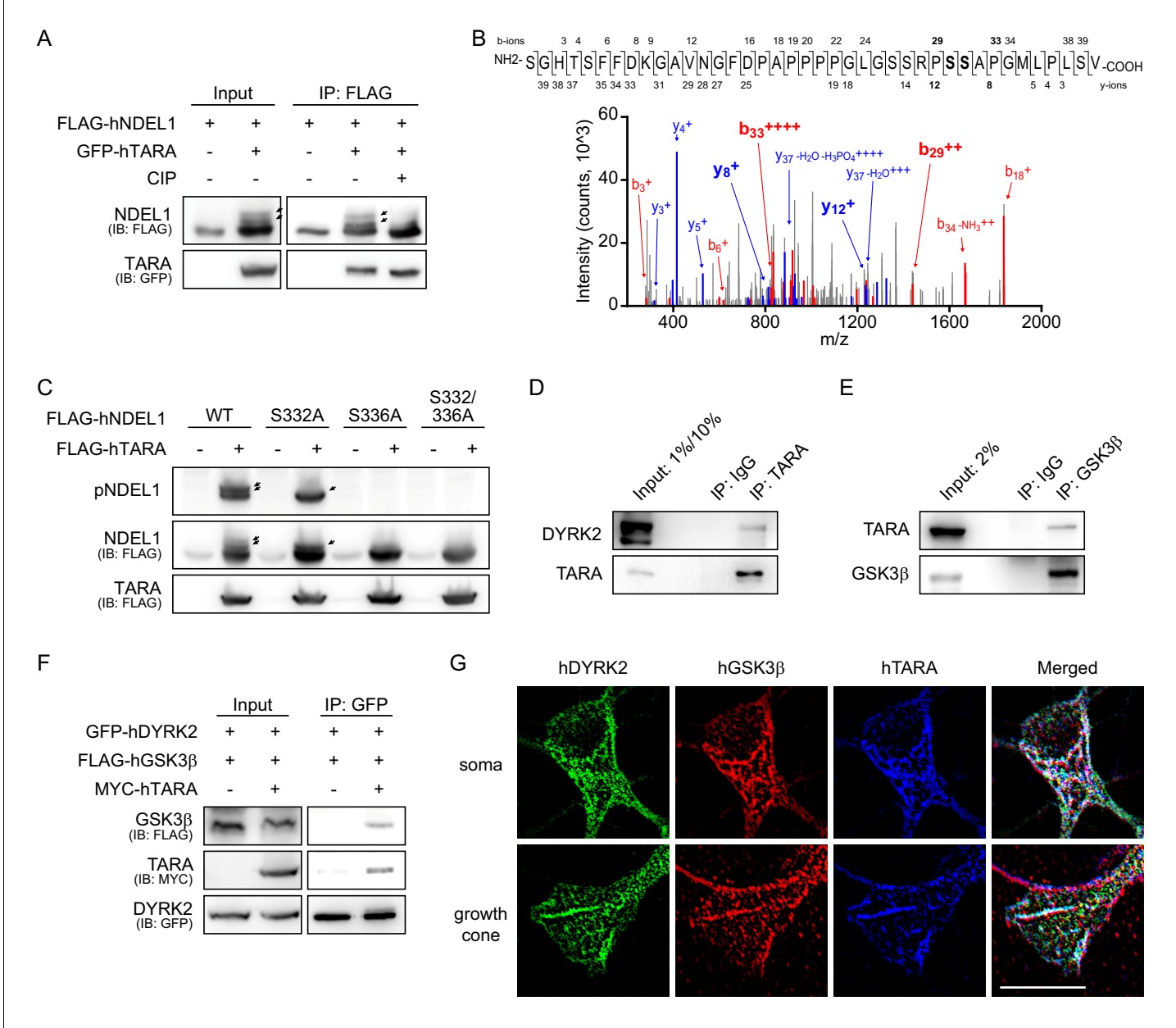

**Figure 4.** TARA recruits DYRK2 and GSK3β to induce sequential phosphorylation of NDEL1 S336/S332. (**A**) *In vitro* phosphatase assay. Calf intestinal alkaline phosphatase (CIP) was treated *in vitro* after IP of FLAG-tagged NDEL1. Additional bands of NDEL1 disappeared upon CIP treatment indicating that these additional bands are caused by the multiple phosphorylation. (**B**) MS/MS spectrum for the phosphorylated fragments of NDEL1 peptide including S332 and S336 residues. NDEL1 proteins from HEK293 cells over-expressing NDEL1 and TARA were subjected to LC-MS/MS analysis. The sequence of the peptide (aa 306–345) and all detected fragment ions are shown above. The b- and y-ions annotated in the spectrum include the sizes of $y_{12}$ and $b_{33}$ ions indicative of a single phosphorylation at either S335 or S336. (**C**) TARA-induced NDEL1 phosphorylation S336 and S332. TARA increased sequential phosphorylation of NDEL1, first at S336 followed by S332. (**D**) The protein-protein interaction between DYRK2 and TARA. Endogenous DYRK2 was co-precipitated by IP of endogenous TARA from HEK293 cell lysates. Rabbit IgG was used as a negative control. At input lanes, 1% and 10% of lysates were loaded for anti-DYRK2 and anti-TARA blots, respectively. (**E**) The protein-protein interaction between GSK3β and TARA. Endogenous TARA was co-precipitated by IP of endogenous GSK3β proteins from HEK293 cell lysates. Mouse IgG was used as a negative control. (**F**) Co-IP among DYRK2, GSK3β, and TARA. Over-expression of TARA increased an amount of GSK3β proteins co-precipitated by IP of DYRK2, implying that TARA scaffolds these kinases to form a DYRK2-GSK3β-TARA tripartite complex. (**G**) Colocalization of DYRK2, GSK3β, and TARA in mouse hippocampal neurons. GFP-hDYRK2, FLAG-hGSK3β, and MYC-hTARA colocalized at the soma (above) and the growth cone (bottom) regions. See also *Figure 4—figure supplements 1* and *2*. The following figure supplement is available for *Figure 4*.

The online version of this article includes the following figure supplement(s) for figure 4:

**Figure supplement 1.** Figure related to *Figure 4*.

*Figure 4 continued on next page*

Figure 4 continued

**Figure supplement 2.** Table for the list of NDEL1 PTMs identified from LC-MS/MS analysis of lysates of HEK293 cells transfected with FLAG-hNDEL1 and GFP-hTARA.

NDEL1[S332/336A], significantly rescued the impaired F-actin dynamics indicating its phosphorylation dependency. Reduced amount of fluorescence recovery can be interpreted as either by the more stable F-actin structures or by the less newly formed F-actin. Since NDEL1 S336/S332 phosphorylation had a minimal effect on an insoluble fraction of F-actin fraction, it is more likely that suppression of NDEL1 phosphorylation reduced F-actin formation.

On the other hand, we tested microtubule dynamics by FRAP assay with mCherry-α-tubulin (*Figure 6—figure supplement 1E–L*). Co-expression of either NDEL1[WT] or NDEL1[S332/336A] with NDEL1 shRNA enhanced microtubule dynamics and the effect was not distinguishable. While NDEL1 regulates microtubule dynamics in collaboration with dynein-LIS1 (*Chansard et al., 2011*; *Niethammer et al., 2000*; *Yamada et al., 2010*), interactions between NDEL1 and either dynein intermediate chain (DYNC1I1) or LIS1 was not significantly affected by the phosphorylations (*Figure 6—figure supplement 2*). Furthermore, when we tested lysosomal trafficking in neurons as a readout for NDEL1-LIS1-dynein activity (*Klinman and Holzbaur, 2015*; *Pandey and Smith, 2011*), both NDEL1[WT] and NDEL1[S332/336A] increased a retrograde movement of GFP-LAMP1 labeled lysosomes (*Figure 6—figure supplement 3*).

Taken together, our results indicate that NDEL1 phosphorylation at S336 and S332 up-regulates F-actin dynamics, at the growth cone of extending neurites, which is likely to underlie neuronal morphogenesis.

## Discussion

In this study, we have identified a novel mechanism underlying neuronal morphogenesis that sequential phosphorylation of NDEL1 at S336 and S332 mediated by the TARA-DYRK2-GSK3β complex promotes the modulation of F-actin dynamics to impact this process (*Figure 7*).

The identification of the TARA-DYRK2-GSK3β signaling module may allow the discovery of novel mechanisms and players in the neuronal morphogenesis. The concerted action of DYRK2 and GSK3 on substrates, such as eIF2Bε, tau, CRMP4, DCX, c-Jun, and c-Myc (*Cole et al., 2006*; *Nishi and Lin, 2005*; *Slepak et al., 2012*; *Taira et al., 2012*; *Tanaka et al., 2012*; *Weiss et al., 2013*; *Woods et al., 2001*), has implicated these kinases in cell cycle, neurite outgrowth, neuronal migration, microtubule regulation, and apoptosis. In these processes, DYRK2 primes the substrate followed by the preferential action of GSK3 at a nearby residue. Likewise, TARA also functions in cell mitosis, migration, and neurite outgrowth (*Bradshaw et al., 2017*; *Hong et al., 2016*; *Zhu et al., 2012*), suggesting that it may co-operate with these kinases to modulate these cellular functions. Here, we newly identified TARA directly binds to DYRK2 and GSK3β and recruits them to NDEL1 for phosphorylation at S336 and S332, respectively. NDEL1 S336 is the priming site; once mutated, phosphorylation at S332 no longer occurs. Furthermore, enhancing TARA expression augments NDEL1 phosphorylation by DYRK2 and GSK3β. Taken together, our results indicate that TARA acts as a molecular scaffold to functionally link these two kinases to NDEL1. Of particular note, TARA and CRMP1, another substrate of GSK3, have been identified in insoluble aggregates present in brain samples of schizophrenia patients (*Bader et al., 2012*), hinting at the potential involvement of TARA-DYRK2-GSK3β signaling module in the associated disease pathogenesis.

NDEL1 S336/S332 phosphorylation may highlight the functional difference between NDEL1 and its paralog NDE1. The C-terminal of NDEL1 (aa 191–345) contains multiple phosphorylation sites that are targeted by Aurora A and CDK1/CDK5 (*Mori et al., 2007*; *Niethammer et al., 2000*). Here, we identified and characterized two novel sites, S336 and S332, that are also located in the C-terminus. Interestingly, these TARA-DYRK2-GSK3β responsible sites, but not other phosphorylation sites, are absent in NDE1, a paralog of NDEL1 (*Bradshaw et al., 2013*; *Mori et al., 2007*; *Niethammer et al., 2000*; *Shmueli et al., 2010*). Indeed, the function of NDEL1 S336/S332 phosphorylation in axon/dendrite outgrowth is not shared by NDE1 (*Figure 2—figure supplement 1L–O*). Thus, we expect that other NDEL1-specific functions are also regulated by S336/S332

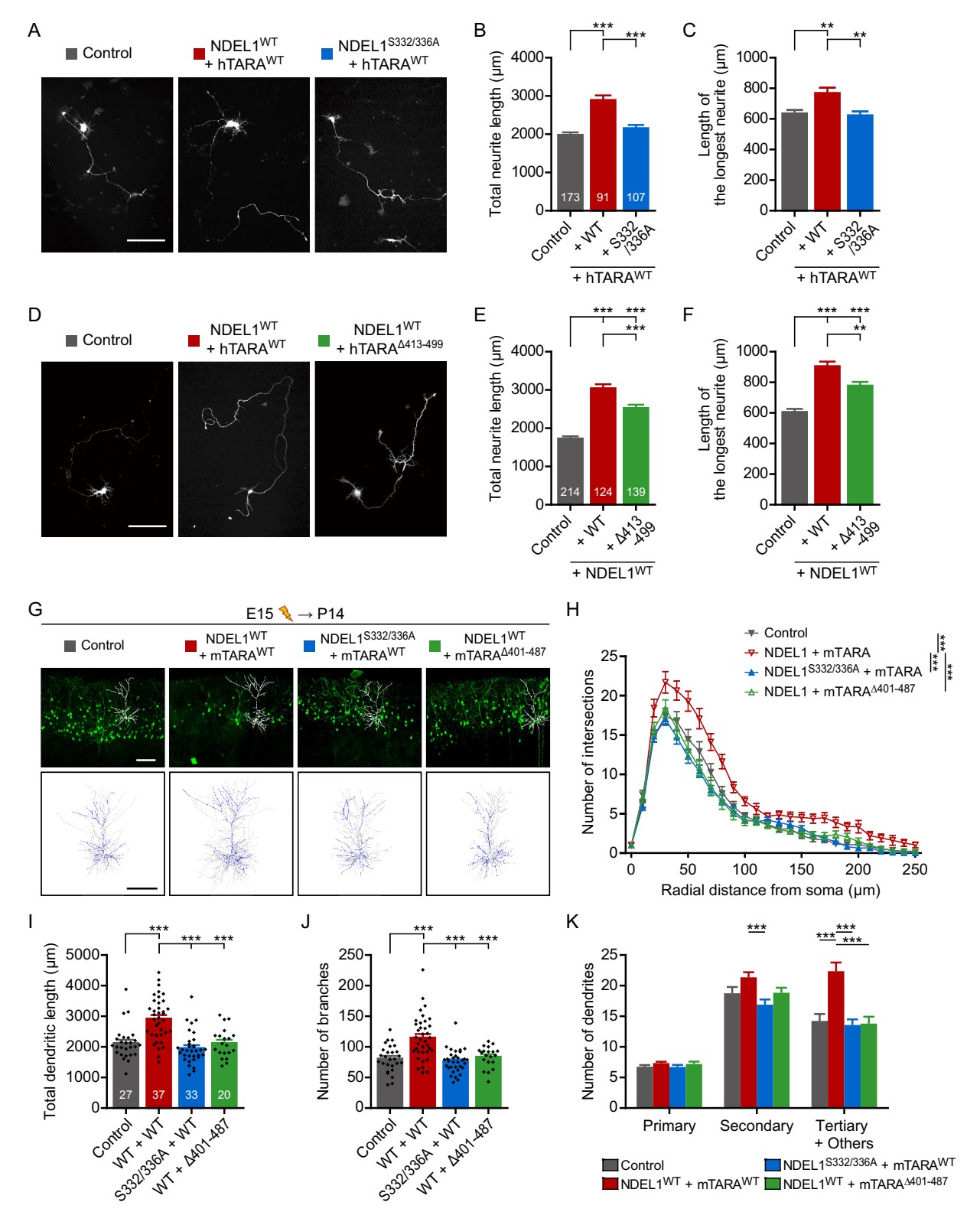

**Figure 5.** NDEL1 S336/S332 phosphorylation induced by TARA increases neurite outgrowth and dendritic arborization. (A–C) Induction of NDEL1 S336/S332 phosphorylation by TARA increased neurite outgrowth. Each DNA construct was transfected at DIV 2 and neurites of transfected neurons were analyzed at DIV 5. (A) Representative images of transfected neurons. The total neurite length (B) and the longest neurite length (C) were measured by using ImageJ software. (D–F) TARA$^{\Delta413-499}$ could not increase neurite outgrowth. Each DNA construct was transfected at DIV 2 and neurites of

*Figure 5 continued on next page*

*Figure 5 continued*

transfected neurons were analyzed at DIV 5. (D) Representative images of transfected neurons. The total neurite length (E) and the longest neurite length (F) were measured by using ImageJ software. (G–K) Up-regulated NDEL1 S336/S332 phosphorylation increased dendritic arborization of layer II/III pyramidal neurons. All constructs were electroporated *in utero* to E15 mouse brain and P14 brain was subjected for analysis. (G) Representative images of the brain slices with the tracked neuron (above) and the overlapped dendritic structures of five independent neurons (bottom). Sholl analysis plots (H), the total length of dendrites (I), the total number of branches (J), and the number of primary/secondary/tertiary dendrites (K) were analyzed by using Simple neurite tracer plug-in of ImageJ software. Each n number is shown at the bottom of the bar of the graph. Scale bars represent 100 μm. All results are presented as means ± SEM. **p<0.01 and ***p<0.001 from one-way ANOVA for (B), (C), (E), (F), (I), and (J) and two-way ANOVA for (H) and (K). All neurite outgrowth experiments for (A–C) and (D–F) were independently repeated for at least three times. All brain samples for each group of (G–K) were collected from offspring of at least three independent *in utero* electroporation surgeries. See also *Figure 5—figure supplement 1*, *Figure 5—video 1*, and *Figure 5—source data 1* and *2*.

The online version of this article includes the following video, source data, and figure supplement(s) for figure 5:

**Source data 1.** Source data for axon/dendrite outgrowth of NDEL1 and TARA over-expression groups.
**Source data 2.** Source data for dendritic arborization of NDEL1 and TARA over-expression groups.
**Figure supplement 1.** Figure for additional data on axon/dendrite outgrowth and dendritic arborization by TARA co-expression.
**Figure 5—video 1.** Video for 3D reconstructions of the dendritic structures of representative neurons, related to *Figure 5G*.
https://elifesciences.org/articles/50850#fig5video1

phosphorylation. On the other hand, recently, Okamoto et al. determined that suppression of DISC1-binding zinc finger protein (DBZ) enhances the dual phosphorylation of NDEL1 at T219 and S251 by CDK5 and Aurora A, respectively (*Okamoto et al., 2015*). This dual phosphorylation inhibits LIS1-DISC1 transport to the neurite tips and thus microtubule elongation, thereby inhibiting radial migration of neurons. Interestingly, phosphorylation by CDK5 alone enhances neuronal migration via increasing NDEL1 binding to katanin p60 (*Toyo-Oka et al., 2005*). Similarly, phosphorylation at S251 by Aurora A alone increases radial migration (*Takitoh et al., 2012*). These results indicate that dual phosphorylation of NDEL1 can have distinct effects to those elicited by single phosphorylation, and similar regulation may also exist regarding NDEL1 S336/S332 phosphorylation and TARA-DYRK2-GSK3β signaling.

The actin-microtubule crosstalk is critical for neuronal morphogenesis (*Coles and Bradke, 2015*; *Dong et al., 2015*; *Pacheco and Gallo, 2016*; *Rodriguez et al., 2003*). TARA directly associates to F-actin (*Seipel et al., 2001*) while NDEL1 regulates actin, microtubules, and intermediate filaments (*Hong et al., 2016*; *Nguyen et al., 2004*; *Niethammer et al., 2000*; *Shim et al., 2008*; *Toth et al., 2014*). We previously showed that TARA recruits NDEL1 toward the cell periphery and together they up-regulate local F-actin levels (*Hong et al., 2016*). Here, we showed that TARA-DYRK2-GSK3β axis promotes NDEL1 S336/S332 phosphorylation to modulate F-actin dynamics (*Figure 6*). We also observed this phosphorylation to be prominent in the neuronal growth cone where they colocalize with both F-actin and microtubules (*Figure 1G*). Some proteins involved in the actin-microtubule crosstalk, such as EB3, Drebrin, and CLIP-170, modulate F-actin dynamics while bound to microtubules (*Jaworski et al., 2009*; *Lewkowicz et al., 2008*; *Mikati et al., 2013*). Likewise, we postulate that NDEL1 may be associated with microtubule and simultaneously up-regulate F-actin dynamics, enhancing actin-microtubule crosstalk at growth cones. Also, we cannot exclude the possibility that TARA-mediated phosphorylation modulates the association of NDEL1 with microtubule. Additional investigations in this regard are required for further understanding how NDEL1 modulate the actin-microtubule crosstalk.

In summary, we have identified the sequential phosphorylation of NDEL1 at S336 and S332 mediated by the TARA-DYRK2-GSK3β complex as a novel regulatory mechanism for neuronal morphogenesis through modulating F-actin dynamics. Harnessing the biology of NDEL1 S336/S332 phosphorylation or the TARA-DYRK2-GSK3β signaling module will provide new insights toward the discovery of novel components and pathways that are pertinent to brain development and neurodevelopmental disorders.

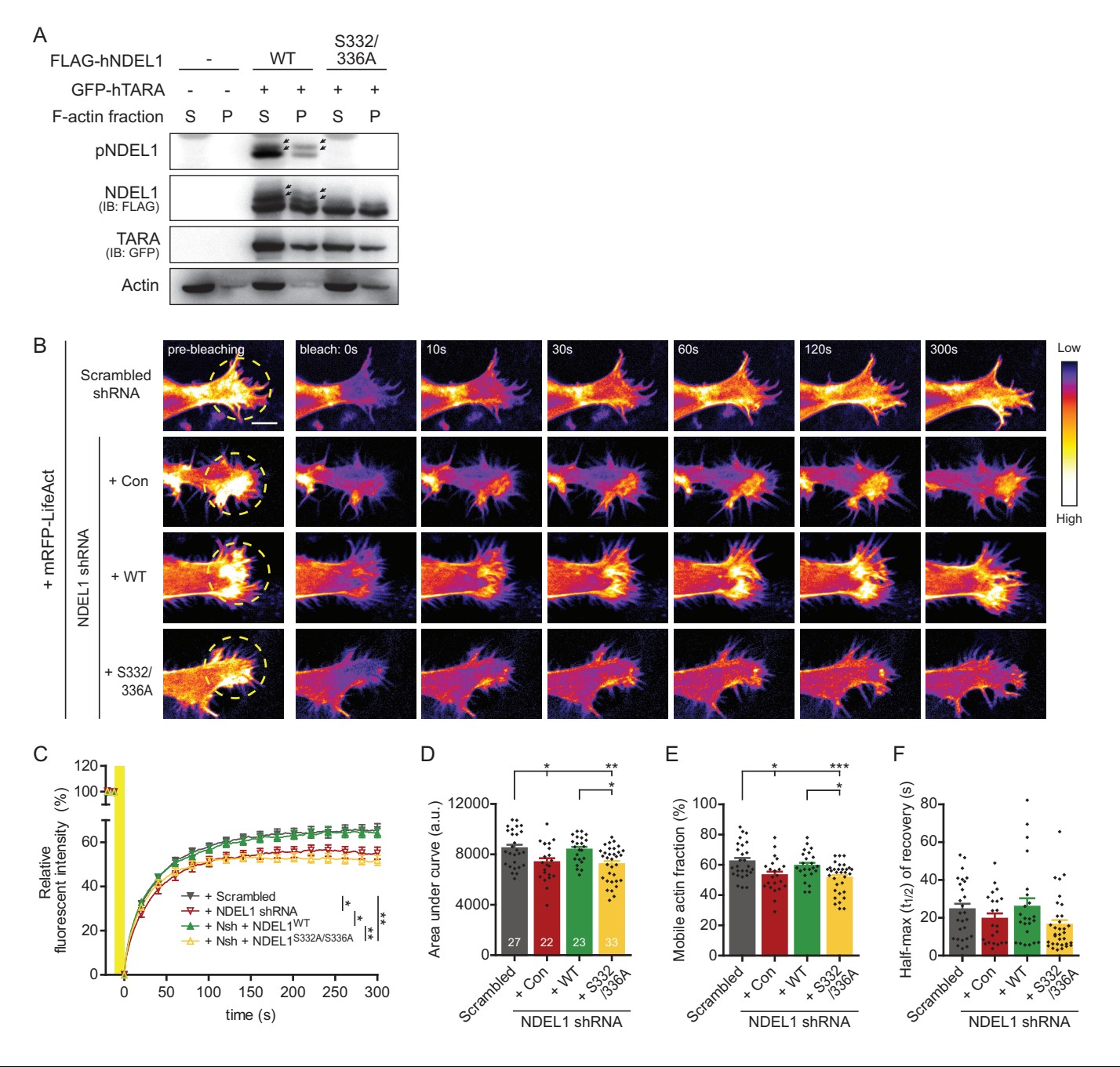

**Figure 6.** Phosphorylation of NDEL1 S336/S332 modulates F-actin dynamics. (**A**) F-actin fractionation assay of NDEL1 and TARA. NDEL1 S336/S332 phosphorylation induced by TARA was observed at both G-actin in the supernatant fraction (S) and F-actin in pellet fraction (P). (**B–F**) Decreased F-actin dynamics by suppression of NDEL1 S336/S332 phosphorylation. (**B**) Representative time-lapse images of FRAP assay to measure F-actin dynamics at differentiating SH-SY5Y cells expressing RFP-LifeAct. All of NDEL1 over-expressing constructs here contain an shRNA-resistant mutation. A yellow-dashed circle indicates region-of-interest used for bleaching. Bleaching was given by stimulating with 10% 568 nm laser for 10 s. (**C**) Time-dependent fluorescence recovery graph. Comparisons of the area under FRAP curves (**D**), the percentage of mobile F-actin fraction calculated by the amount of eventual fluorescence recovery (**E**), and the average half-max ($t_{1/2}$) of RFP-LifeAct fluorescence recovery (**F**). NDEL1 knockdown cells had decreased fluorescence recovery meaning more immobile fraction of F-actin and could not be rescued by NDEL1[S332/336A] implying its phosphorylation dependency. Each n number is shown at the bottom of the bar of the graph. The scale bar at (**B**) represents 10 μm. All results are presented as means ± SEM. *p<0.05, **p<0.01, and ***p<0.001 from two-way ANOVA for (**C**) and one-way ANOVA for (**D–F**). See also *Figure 6—figure supplements 1*, *2* and *3*, *Figure 6—video 1*, and *Figure 6—source data 1*.

The online version of this article includes the following video, source data, and figure supplement(s) for figure 6:

*Figure 6 continued on next page*

*Figure 6 continued*

**Source data 1.** Source data for F-actin FRAP assay of NDEL1 knockdown and rescue groups.
**Figure supplement 1.** Figure of raw data for F-actin FRAP assay plots and results for Microtubule FRAP assay.
**Figure supplement 2.** Figure for additional data on the effect of S336/S332 phosphorylation on the interaction of NDEL1 with DYNC1I1 and LIS1.
**Figure supplement 3.** Figure for additional data on the effect of S336/S332 phosphorylation on the lysosomal trafficking.
**Figure 6—video 1.** Video for the time-lapse imaging series of the FRAP assay for representative cells, related to *Figure 6B*.
https://elifesciences.org/articles/50850#fig6video1

## Materials and methods

### Animals
Pregnant Sprague Dawley (SD) rat and ICR mice were purchased from Hyochang Science (Daegu, South Korea) and used for primary hippocampal neuron culture, brain lysate preparation, and *in utero* electroporation surgery. All animal procedures were approved by the Institutional Animal Care and Use Committee (IACUC) of Pohang University of Science and Technology (POSTECH-2019–0024 and POSTECH-2019–0025). All experiments were carried out in accordance with the approved guidelines.

### Antibodies and plasmids
Anti-NDEL1 rabbit polyclonal antibody (Cat# 17262–1-AP, RRID:AB_2235821) was purchased from Proteintech Group (Rosemont, IL, USA). Anti-TARA rabbit polyclonal antibody (Cat# PA5-29092,

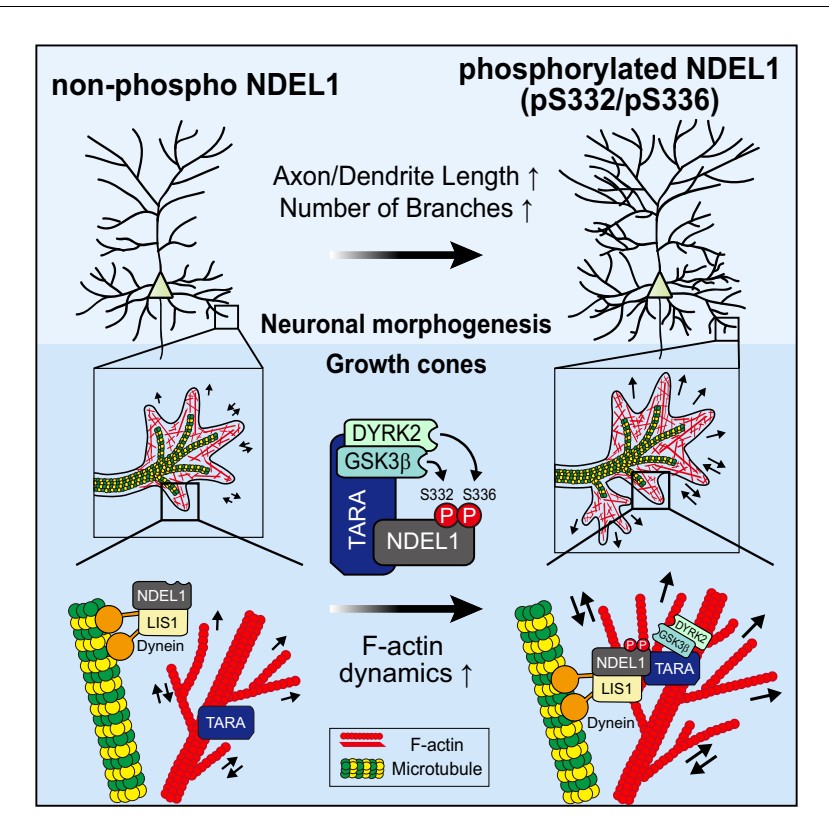

**Figure 7.** A model for the mechanism by which NDEL1 S336/S332 phosphorylation regulates neuronal morphogenesis. The phosphorylation of NDEL1 at S336 by DYRK2 primes S332 phosphorylation by GSK3β. TARA mediates this process by recruiting DYRK2 and GSK3β to NDEL1 and forming a tripartite complex in association with F-actin. The phosphorylated NDEL1 enhances F-actin dynamics at the interface with microtubule cytoskeleton in growth cones, thereby facilitating axon/dendrite length and neuronal arborization.

RRID:AB_2546568) was purchased from Thermo Fisher Scientific (Waltham, MA, USA). The NDEL1 S336/S332 phosphorylation-specific antibody (anti-pNDEL1) was generated and purified from rabbit blood serum after repetitive immunizations with KLH-conjugated phospho-NDEL1 peptide (KLH-C-GLGSpSRPSpSAPG, AbClon). Anti-GSK-3beta mouse monoclonal (Cat# 9832, RRID:AB_10839406, Cell Signaling Technology, Danvers, MA, USA) and anti-DYRK2 mouse monoclonal (Cat# MA5-24269, RRID:AB_2606267, Thermo Fisher Scientific) were used for IP and immunoblotting experiments. Anti-FLAG rabbit polyclonal and mouse monoclonal (Cat# F7425, RRID:AB_439687 and Cat# F1804, RRID:AB_262044, respectively, Sigma-Aldrich, St. Louis, MO, USA), anti-GFP rabbit polyclonal (Cat# A-11122, RRID:AB_221569, Molecular Probes, Eugene, OR, USA), anti-GFP mouse monoclonal (Cat# sc-9996, RRID:AB_627695, Santa Cruz Biotechnology, Santa Cruz, CA, USA), anti-α-tubulin mouse monoclonal (Cat# sc-32293, RRID:AB_628412, Santa Cruz Biotechnology, and Cat# 66031–1-Ig, RRID:AB_11042766, Proteintech Group), and anti-c-Myc mouse monoclonal (Cat# sc-40, RRID:AB_627268, Santa Cruz Biotechnology) were used for immunoblotting, IP, and immunostaining experiments. Anti-actin goat polyclonal (Cat# sc-1616, RRID:AB_630836, Santa Cruz Biotechnology) and PhosphoSerine Antibody Q5 (37430, Qiagen, Germantown, MD, USA) were used for immunoblotting. As a negative control for immunostaining and IP, normal rabbit IgG (Cat# sc-2027, RRID:AB_737197, Santa Cruz Biotechnology, and Cat# ab37415, RRID:AB_2631996, Abcam, Cambridge, UK) and normal mouse IgG (Cat# sc-2025, RRID:AB_737182, Santa Cruz Biotechnology) were used. For immunoblotting, HRP-conjugated sheep anti-mouse IgG (Cat# NA931, RRID:AB_772210, GE Healthcare, Buckinghamshire, UK) and donkey anti-rabbit IgG (Cat# NA934, RRID:AB_772206, GE Healthcare) were used as secondary antibodies. For immunoblotting of IP, VeriBlot for IP Detection Reagent (HRP) (Cat# ab131366, Abcam) was also used as secondary antibody. For immunostaining, Alexa Fluor 488, Alexa Fluor 568, or Flamma 648 conjugated goat anti-rabbit IgG (Cat# A-11008, RRID:AB_143165 and Cat# A-11011, RRID:AB_143157, Molecular Probes and Cat# RSA1261, Bio-Acts, Incheon, South Korea) and Alexa Fluor 488 or 568 conjugated goat anti-mouse antibodies (Cat# A-11004, RRID:AB_141371 and Cat# A-21235, RRID:AB_141693, Molecular Probes) were used as secondary antibodies.

All constructs for human *NDEL1*, human *TRIOBP* (isoform 1, hTARA), and mouse *Triobp* (isoform 1, mTARA) were prepared by cloning into pFLAG-CMV2 (Sigma-Aldrich), pEGFP-C3 (Clontech, Mountain View, CA, USA), and pcDNA3.1/myc-His (Invitrogen). Constructs for human *DYRK2* (hDYRK2) and human *GSK3B* (hGSK3β) were cloned into pEGFP-C3 and pFLAG-CMV2. The construct for mouse *Nde1* (NCBI nucleotide ID: NM_023317.2) was prepared by cloning into pCIG2-mRFP vector. All shRNA constructs were designed by cloning 19–21 nt of core sequences combined with TTCAAGAGA as the loop sequence into pLentiLox3.7 vector as described previously (*Brummelkamp et al., 2002*; *Hong et al., 2016*). Core sequences of NDEL1 shRNA, responsible to both human *NDEL1* and mouse and rat *Ndel1*, and hTARA shRNA were GCAGGTCTCAGTGTTA-GAA and GCTGACAGATTCAAGTCTCAA, respectively, as we described previously (*Hong et al., 2016*; *Nguyen et al., 2004*). The core sequence of control scrambled shRNA was CTACCGTTGTA TAGGTG. All constructs for *in utero* electroporation were cloned into pCIG2-mRFP, pCIG2-EGFP, and pUBC vectors. All expression constructs for human kinome library were cloned into pEZYmyc-His (Addgene plasmid # 18701) and pEZYflag (Addgene plasmid # 18700) Gateway destination vectors, which were gifts from Yu-Zhu Zhang (*Guo et al., 2008*). hCas9_D10A (Addgene plasmid # 41816), a construct used to generate NDEL1 S332/336A KI, was a gift from George Church (*Mali et al., 2013*). Core sequences of two guide RNAs targeting mouse *Ndel1* exon nine were TC TTCTCGCCGTAGTGCCGT and ATTGATATCGCGCAGAGTCC.

## Cell culture and transfection

HEK293 and NIH3T3 cells were cultured in DMEM (HyClone, South Logan, UT, USA) supplemented with 10% (v/v) fetal bovine serum (FBS) (Gibco, Gaithersburg, MD, USA) and 1% penicillin/streptomycin (Gibco). SH-SY5Y cells were grown in MEM supplemented with 10% (v/v) FBS and 1% penicillin/streptomycin and were differentiated by treatment of 10 μM all-trans retinoic acid in MEM supplemented with 2% FBS for more than 3 days. All cell lines were authenticated using STR profiling method and were tested negative for mycoplasma contamination. All cells were transfected by using transfection reagent either VivaMagic (Vivagen, Seongnam, South Korea) or Lipofectamine 2000 (Thermo Fisher Scientific).

Primary cultures of hippocampal neurons were established by isolating E18 SD rat embryo or E15 ICR mouse embryo hippocampal tissues in HBSS (Gibco) and dissociating tissues in 0.25% trypsin (Sigma-Aldrich) and 0.1% DNase I (Sigma-Aldrich) for 10 min at 37°C. Cells were resuspended in Neurobasal medium (Gibco) supplemented with 10 mM HEPES pH 7.4% and 10% (v/v) horse serum for final cell concentration being $4.0 \times 10^5$ cells/mL and plated on glass coverslips pre-coated with poly-D-lysine and laminin. After 2 hr of plating, cell medium was replaced to Neurobasal medium containing 2 mM glutamine, 2% (v/v) B27 supplement (Gibco), and 1% (v/v) penicillin/streptomycin. The neurons were transfected 9 hr or 48 hr after plating with Lipofectamine 2000 and medium was replaced to the culture medium 2 hr after transfection.

## F-actin fractionation analysis

Transfected HEK293 cells were lysed in actin fractionation buffer (10 mM Tris pH 7.4, 2 mM MgCl$_2$, 1% Triton X-100, 0.2 mM DTT, and 15% glycerol). After homogenization, lysates were subjected to centrifugation at 3,000 rpm for 1 min to remove cell debris. Supernatant fraction (G-actin) and pellet fraction (F-actin) were separated by centrifugation at 100,000 g (4°C) for 1 hr. Fractions were analyzed by immunoblotting.

## Human kinome library screening

The Center for Cancer Systems Biology (Dana Farber Cancer Institute)-Broad Human Kinase ORF collection plasmid kit (*Johannessen et al., 2010*; *Yang et al., 2011*)[2, 5] was a gift from William Hahn and David Root (Addgene kit # 1000000014). Each kinase ORF in pDONR-223 vector was cloned into pEZYmyc-His or pEZYflag Gateway destination vectors via LR Clonase II Plus enzyme (Thermo Fisher Scientific) reaction for 4 hr at room temperature followed by transformation into DH5α competent cells and selection from ampicillin-containing LB agar plate. Each kinase ORF expressing clone was confirmed by sequencing analysis.

For screening responsible kinases for NDEL1 S336/S332 phosphorylation, FLAG-NDEL1 plasmid and the expressing clone plasmid were transfected into HEK293 cells and incubated for 48 hr. pEGFP-C3 plasmid and MYC-hTARA construct were transfected with FLAG-NDEL1 to be used as a negative control and a positive control, respectively. Cells were lysed into 1X ELB lysis buffer (50 mM Tris pH 8.0, 250 mM NaCl, 5 mM EDTA, 0.1% NP-40) supplemented with 2 mM NaPPi, 10 mM NaF, 2 mM Na$_3$VO$_4$, 1 mM DTT, and protease inhibitor cocktail (Roche, Mannheim, Germany). The lysates were subjected to immunoblotting with FLAG antibody. The candidate kinases were selected by the increment of NDEL1 phosphorylation evidenced by the band shift.

## Immunoblot assay and immunoprecipitation

Transfected HEK293 cells were lysed in 1X ELB lysis buffer supplemented with 2 mM NaPPi, 10 mM NaF, 2 mM Na$_3$VO$_4$, 1 mM DTT, and protease inhibitor cocktail (Roche). Mouse brain tissues were isolated from anesthetized and perfused mice followed by homogenization and lysis in 1X modified RIPA lysis buffer (50 mM Tris pH 7.5, 150 mM NaCl, 1% NP-40, 5 mM EDTA, 1% Triton X-100, 0.5% sodium deoxycholate) supplemented with 2 mM NaPPi, 10 mM NaF, 2 mM Na$_3$VO$_4$, 1 mM DTT, protease inhibitor cocktail (Roche) and 10 U/mL Benzonase nuclease (Sigma-Aldrich). For immunoblotting analysis, proteins were denatured by mixing lysates with 5X SDS sample buffer (2% SDS, 60 mM Tris pH 6.8, 24% glycerol, and 0.1% bromophenol blue with 5% β-mercaptoethanol) and incubating at 95°C for 10 min. Proteins were separated by SDS-PAGE with 9% polyacrylamide gel and transferred to PVDF membrane (Millipore, Billerica, MA, USA). Membranes were blocked with 5% skim milk or 4% bovine serum albumin (BSA) in Tris-buffered saline (20 mM Tris pH 8.0, and 137.5 mM NaCl) with 0.25% Tween20 (TBST) for 30 min and incubated with primary antibodies at 4°C for more than 6 hr and HRP-conjugated secondary antibodies at room temperature for more than 2 hr. Protein signals were detected by ECL solutions (BioRad, Hercules, CA, USA). For IP, lysates were incubated with 1–5 μg of antibody at 4°C for more than 6 hr with constant rotation. Protein-A agarose beads (Roche) washed three times with lysis buffer were mixed with IPed lysates and incubated at 4°C for 2 hr or overnight with constant rotation. Beads were collected by centrifugation, washed three times, and mixed with SDS sample buffer for immunoblotting analysis.

## Immunocytochemistry and immunohistochemistry

For immunocytochemistry, cells were fixed with 4% paraformaldehyde in PBS or 4% paraformaldehyde and 4% sucrose in PBS for 20 min and washed with PBS for three times. Cells were permeabilized with 0.5% Triton X-100 in PBS for 5 min and blocked with 5% goat serum in PBS or 4% BSA in PBS for more than 30 min. For staining proteins, cells were incubated with primary antibodies diluted in the blocking solution for 1 hr at room temperature or overnight at 4°C, washed with PBS for three times, and treated with secondary antibodies diluted in the blocking solution for 1 hr at room temperature.

For sequential immunostaining, cells were incubated with the first primary antibody diluted in the blocking solution for 2 hr followed by two rounds of incubation with Alexa Fluor 488-conjugated secondary antibody in the blocking solution for 1 hr each at room temperature. Cells were washed with PBS for more than three times, incubated with the second primary antibody diluted in the blocking solution for 2 hr at room temperature, and treated with Alexa Fluor 647-conjugated secondary antibody in the blocking solution for 1 hr at room temperature.

For immunohistochemistry, mouse brain slices on slide glass were washed with PBS and additionally fixed with 4% paraformaldehyde in PBS for 20 min. After three times of PBS washing, 0.5% Triton X-100 in PBS was treated for permeabilization for 10 min and 5% goat serum or 5% BSA blocking solution was treated for 1 hr. Primary antibody diluted in blocking solution was treated for overnight at 4°C. After three times of PBS washing, fluorescent-conjugated secondary antibody diluted in blocking solution was treated for 2 hr at room temperature. For sequential staining, it was done as same as immunocytochemistry. Tissue slides were washed with PBS and mounted by using UltraCruz Aqueous Mounting Medium with DAPI (Cat# sc-24941, RRID:AB_10189288, Santa Cruz Biotechnology).

Cell images were acquired by using FV3000 confocal laser scanning microscope (Olympus, Tokyo, Japan) and processed by using ImageJ (Fiji) software (RRID:SCR_002285, National Institute of Health, Bethesda, MD, USA) (*Schindelin et al., 2012*). For quantitation of colocalization between NDEL1, TARA, pNDEL1, and IgG control staining patterns, all images were deconvolved using advanced constrained iterative (CI) algorithm-based deconvolution program of cellSens software (Olympus) and were subjected for Pearson's colocalization coefficient analysis through cellSens.

## In utero electroporation

Pregnant ICR mice at E15 were anesthetized with an intraperitoneal injection of ketamine (75 mg/Kg) (Yuhan Corporation, Seoul, South Korea) and xylazine (11.65 mg/Kg) (Bayer AG, Leverkusen, Germany) in PBS. Coding sequences of target genes in pCIG2 vectors or shRNA sequences in pLL3.7-EGFP and pLL3.7-mRFP vectors were purified by using EndoFree plasmid maxi kit (Qiagen, Germantown, MD, USA). Each DNA solution (2.0 µg/µL) mixed with Fast Green solution (0.001%) was injected into bilateral ventricles of the embryo through pulled microcapillary tube (Drummond Scientific, Broomall, PA, USA). Tweezer-type electrode containing two disc-type electrodes was located with appropriate angle and electric pulses were given as 35 V, 50 ms, five times with 950 ms intervals by using an electroporator (Harvard Apparatus, Holliston, MA, USA). After electroporation, embryos were put back into the mother's abdomen, the incision was sutured, and mice were turned back to their home cage. The mice were sacrificed at E18 or P14 and brains were fixed with 4% paraformaldehyde in PBS for 24 hr, dehydrated with 10% and 30% sucrose in PBS for more than 24 hr, and soaked and frozen in Surgipath FSC22 Clear OCT solution (Leica Biosystems, Richmond, IL, USA). Brain tissue was sectioned by using cryostats (Leica Biosystems) with 100 µm thickness and each section was immediately bound to Superfrost Plus microscope slides (Fisher Scientific). Brain slice images were acquired by using 10x, 20x, and 40x objective lenses of FV3000 confocal laser scanning microscope (Olympus) with Z-stacks of 1 µm intervals.

## In vitro axon/dendrite outgrowth assay

Primary cultured rat hippocampal neurons were subjected to transfection at either 9 or 48 hr after plating. The neurons were fixed by 4% (w/v) paraformaldehyde in PBS for 20 min after 72 or 48 hr after transfection for the knockdown group or the over-expression groups, respectively. Cell images were acquired by using a 40x objective lens of fluorescent microscopy and analyzed by using ImageJ software.

## In vitro phosphatase assay

Transfected HEK293 cells were lysed in 1X ELB lysis buffer and were IPed with FLAG antibody and protein-A agarose beads to enrich FLAG-tagged NDEL1 proteins. After washing and removal of PBS, beads were incubated with 1X in vitro kinase assay buffer (30 mM HEPES pH 7.2, 10 mM MgCl$_2$, and 0.2 mM DTT) and 10 units of calf intestinal alkaline phosphatase (CIP, New England Biolabs, Beverley, MA, USA) at 37°C for 30 min. 5X SDS sample buffer was mixed to stop dephosphorylation activity of CIP and to subject for western blot analysis.

## Liquid Chromatography (LC)-Mass Spectrometry (MS)/MS Analysis

To enrich endogenous NDEL1 proteins from P7 mouse brains, brain lysates with total of 20 mg proteins were IPed with 2 µg of anti-NDEL1 antibody. IPed proteins were reduced with 5 mM dithiothreitol for 0.5 hr at 56°C and alkylated with 20 mM iodoacetamide at room temperature in the dark for 20 min followed by in-bead protein digestion with 1 µg trypsin (Promega) at 37°C overnight. The beads were removed by a short spin and digested peptides were desalted with a C18 spin column (#89870, Thermo Fisher Scientific). The peptides extracted from the IP were analyzed with a nano liquid chromatography (LC) system (Dionex) coupled to a Q-Exactive Plus Orbitrap (Thermo Fisher Scientific). A binary solvent system composed of 0.1% formic acid in water and 0.1% formic acid in acetonitrile was used for all analysis. Peptide fractions were separated on an Ultimate 3000 RSLCnano System with a PepMap 100 C18 LC column (#164535) serving as a loading column followed by a PepMap RSLC C18 (#ES803) analytical column with a flow rate of 0.3 µL/min for 135 min. Full scan mass spectrometry (MS) with a data-dependent MS/MS acquisition was performed in a range from 350 to 2000 m/z. All raw LC-MS/MS data were processed with Proteome Discoverer 2.2 (Thermo Fisher Scientific). Data were filtered to a 1% false discovery rate and searched for dynamic phosphorylation modifications (79.966 Da) at a fragment mass tolerance setting of 0.02 Da.

## Lysosomal trafficking assay

Primary cultured mouse hippocampal neurons were subjected to transfection of GFP-LAMP1 with indicated constructs at DIV 5–7 and imaged at 37°C with supplying 5% CO$_2$ gas by using FV3000 confocal laser scanning microscope. Through FV31S-DT software (Olympus), a 50 µm-length region of interest (ROI) at the axon was determined and recorded for total 3 min with 1 s interval. Generation of kymographs and data analysis were performed by using CellSens and ImageJ using the KymoAnalyzer v1.01 plug-in (Neumann et al., 2017) combined with manual analysis.

## Neuronal dendritic morphology analysis

For neuronal dendritic morphology analysis, mouse embryos were electroporated in utero at E15 and sacrificed at P14. From sectioned brain slices, the somatosensory cortex was located based on mouse brain atlas with hippocampal structure and cortical layer II-IV was distinguished by DAPI staining pattern. Each layer II/III pyramidal neuron in which cell soma is located in the middle of Z-stacks with clearly observable apical dendritic structure was subjected for analysis. Apical and basal dendritic morphologies of each selected neuron were traced out by using Simple Neurite plug-in of ImageJ or Imaris software (Bitplane, Zurich, Switzerland) and total length, the longest length, number of branches, and number of primary/secondary dendrites were measured. For Sholl analysis, the tracing data were subjected to Sholl Analysis plug-in of ImageJ by 10 µm radius step size.

## Time-lapse live imaging with FRAP assay

To test F-actin or microtubule dynamics, differentiated SH-SY5Y cells expressing either RFP-LifeAct or mCherry-α-tubulin were imaged at 37°C with supplying 5% CO$_2$ gas by using FV3000 confocal laser scanning microscope. Through FV31S-DT software, either a 10 µm-radius circular region of interest (ROI) around cellular process tips for F-actin dynamics or 6 µm x 3 µm rectangular ROI at the middle of the cellular process for microtubule dynamics was determined. The ROI was photobleached by scanning with 10% power 568 nm laser and 20 µs/pixel scan speed for total 10 s. For FRAP analysis, five frames were acquired as pre-bleach images followed by bleaching and 150 frames were acquired as post-bleach with each 2 s interval. FRAP results were analyzed by automatically with the easyFRAP-web application (Koulouras et al., 2018) combined with additional manual analysis.

## Statistical analysis

All graphs were presented as the mean ± SEM. Statistical significance of the data was analyzed by two-tailed Student's t-test for comparisons between two groups and one-way or two-way ANOVA followed by Bonferroni's post-hoc test for comparisons among multiple groups.

# Acknowledgements

This work was supported by the Brain Research Program (2015M3C7A1030964 and 2017M3C7A1047875), Advanced Research Center Program (Organelle Network Research Center, 2017R1A5A1015366), and Mid-career Researcher Program (2017R1A2B2009031) funded by Korean National Research Foundation (SKP). This study was also supported in part by the Canadian Institutes of Health Research (MDN) and KBRI basic research program through Korea Brain Research Institute funded by Ministry of Science and ICT (19-BR-02–01, YC).

# Additional information

### Funding

| Funder | Grant reference number | Author |
| --- | --- | --- |
| National Research Foundation of Korea | 2015M3C7A1030964 | Sang Ki Park |
| National Research Foundation of Korea | 2017M3C7A1047875 | Sang Ki Park |
| National Research Foundation of Korea | 2017R1A5A1015366 | Sang Ki Park |
| National Research Foundation of Korea | 2017R1A2B2009031 | Sang Ki Park |
| Canadian Institutes of Health Research | | Minh Dang Nguyen |
| Ministry of Science, ICT and Future Planning | 19-BR-02-01 | Youngshik Choe |

The funders had no role in study design, data collection and interpretation, or the decision to submit the work for publication.

### Author contributions

Youngsik Woo, Conceptualization, Investigation, Methodology; Soo Jeong Kim, Bo Kyoung Suh, Truong Thi My Nhung, Dong Jin Mun, Investigation, Methodology; Yongdo Kwak, Conceptualization, Resources, Investigation, Methodology; Hyun-Jin Jung, Su-Jin Noh, Seunghyun Kim, Ahryoung Lee, Investigation; Ji-Ho Hong, Resources, Methodology; Seung Tae Baek, Methodology; Minh Dang Nguyen, Resources, Funding acquisition; Youngshik Choe, Resources, Funding acquisition, Methodology; Sang Ki Park, Conceptualization, Resources, Supervision, Funding acquisition, Methodology

### Author ORCIDs

Youngsik Woo (iD) https://orcid.org/0000-0002-8308-8532
Bo Kyoung Suh (iD) https://orcid.org/0000-0001-8079-9446
Sang Ki Park (iD) https://orcid.org/0000-0003-1023-7864

### Ethics

Animal experimentation: All animal procedures were approved by the Institutional Animal Care and Use Committee (IACUC) of Pohang University of Science and Technology (POSTECH-2019-0024 and POSTECH-2019-0025). All experiments were carried out in accordance with the approved guidelines. All surgery was performed under ketamine/xylazine cocktail anesthesia, and every effort was made to minimize suffering.

**Decision letter and Author response**
Decision letter https://doi.org/10.7554/eLife.50850.sa1
Author response https://doi.org/10.7554/eLife.50850.sa2

## Additional files

### Supplementary files
- Supplementary file 1. Key resources table.
- Transparent reporting form

### Data availability
All data generated or analysed during this study are included in the manuscript and supporting files. Source data files have been provided for Figures 1, 2, 3, 5, and 6.

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
