## [Decision Letter]

**Acceptance summary:**

Through a comprehensive screen of human kinases and phospho-proteomics analysis of the protein Nuclear distribution element-like 1 (NDEL1), reviewers recognize that the study identifies the DYRK2 kinase as phosphorylating NDEL1 at residue S336 to prime the phosphorylation of NDEL1 at residue S332 by GSK3, with the NDEL1 interaction partner TARA (Trio-associated repeat on actin) scaffolding the DYRK2 and GSK3 kinases to enhance NDEL1 S336/S332 phosphorylation. The result of these posttranslational modifications is to enhance both axonal and dendritic outgrowth and promote their arborization by increasing filamentous actin.

**Decision letter after peer review:**

Thank you for submitting your article "Sequential phosphorylation of NDEL1 by the DYRK2-GSK3β complex is critical for neuronal morphogenesis" for consideration by *eLife*. Your article has been reviewed by Marianne Bronner as the Senior Editor, a Reviewing Editor, and three reviewers. The following individuals involved in review of your submission have agreed to reveal their identity: Yuanyi Feng (Reviewer #1); Deanna Smith (Reviewer #2).

The reviewers have discussed the reviews with one another and the Reviewing Editor has drafted this decision to help you prepare a revised submission.

Summary:

Reviewers recognize the intriguing evidence that TARA acts as a scaffold for NDEL1 and two kinases, DYRK2 and GSK3β, identified using a human kinome screen, and agree that the *in vitro* data supporting the model that DYRK2 phosphorylation NDEL1 S336 primes phosphorylation by GSK3β at S332, and that phosphorylation impacts filamentous actin but not microtubules. However, there are two major concerns raised that reviewers would like to see addressed before the work can move forward at *eLife*. First, all reviewers point out that the work is predominantly limited to tagged proteins, so whether the interactions described also occur with endogenous proteins remains a question. Second, all reviewers request genetic evidence for this interaction other than overexpression, or at the minimum, the protein levels of NDEL1 or pNDEL1 (in the shRNA – overexpression rescue experiments) need to be better controlled for physiological levels. NDEL1 may bind many molecules, data from KD or expression by the CAG promotor are very difficult to interpret unless they are expressed at the right levels (i.e. either null or native).

Essential revisions:

1) Authors should look at other known NDEL1 interactions, and to determine whether the interactions described in this paper also occur with endogenous proteins.

2) Reviewers request genetic evidence, ideally knock in of the non-phosphorylatable or pseudo-phosphorylated residue to demonstrate in vivo evidence of the requirement of these events for neuronal morphogenesis. If not possible, at the minimum, the protein levels of NDEL1 or pNDEL1 (in the shRNA – overexpression rescue experiments) should be controlled to ensure physiological relevance.

3) Is Ndel1 expressed and KD in neuronal (as opposed to glial) progenitors? To understand the function of pNDEL1 phosphorylation, it is necessary to understand cell type (progenitor/neuron/glia) and development stage-specificity of the S332/S336 phosphorylation. However, the phospho antibody also recognizes the native NDEL1, making it difficult to tease this out.

4) The ideal experiment would be to demonstrate reduction of NDEL1 phosphorylation following kinase knockdown (in this paper, only kinase overexpression is performed). However, since this is probably quite challenging, the authors should at least (a) quantify the increase in pNDEL1 signal following kinase overexpression and (b) try to demonstrate the presence of this phosphorylation in another way (phos-Tag gel, MS?). It is not entirely clear to me why the FLAG-NDEL1 shows such a dramatic shift in the kinome experiment (Figure 1—figure supplement 1), but not endogenous NDEL1.

[Editors' note: further revisions were requested prior to acceptance, as described below.]

Thank you for resubmitting your work entitled "Sequential phosphorylation of NDEL1 by the DYRK2-GSK3β complex is critical for neuronal morphogenesis" for further consideration by *eLife*. Your revised article has been evaluated by Marianne Bronner (Senior Editor) and Joseph Gleeson (Reviewing Editor).

The manuscript has been improved but there are some remaining issues that need to be addressed before acceptance, as outlined below. The authors provided thorough responses to the critiques and added high quality new data to address many questions raised by reviewers. The new actin-based mechanism shown by this study appears independent of NDEL1's brain partners, and the developmental analyses of the revised manuscript remain partially unclear due to the lack of specificity of the experiments. The remaining issues could be addressed experimentally or through better description of the possible interpretation or limitations of the approaches applied.

1) Authors acknowledged that the NDEL1 promoter is very weak. In this case, it is concerning to use even the UBC promoter, as there was no information to justify the level of expression delivered by electroporation. Moreover, the effect of loss of NDEL1/pNDEL1 function on neuronal morphology was assessed by knockdown electroporation at E15, but it is not clear how effect knockdown or replacement worked, or the cell types affected (i.e. NPC, glial progenitor, etc).

2) Reviewers remain unconvinced that there is no impact on MTs. The microtubule co-sedimentation assay (Author response image 6) suggests that in the presence of overexpressed GFP-TARA, there is more flag-pNDEL1in the soluble fraction than in the absence overexpressed GFP-TARA. This suggests that the interaction between TARA and NDEL1 might impact the interaction of NDEL1 and MTs.

3) The lysosome trafficking study did not include NDEL1 alone. It is possible that TARA might impact the effect of NDEL1 with respect to organelle trafficking. For example, NDEL1 alone might increase retrograde organelles more than NDEL1S332/336A regardless of TARA, or both WT and mutant NDEL1 could impact transport more in the absence or TARA. This is relevant to the interesting model shown in Figure 7 and may suggest that TARA and the kinases regulate an interplay between F-actin- and MT- associated NDEL1.

4) Please explain the mass spec data from brain (Figure 1D) and HEK293 cells overexpressing NDEL1 and TARA (Figure 4B). From these figures it seems to indicate that S332 is phosphorylated in brain but not in the HEK293 cells. Is that correct?

5) The SSSSC splice variant of NDE1 could potentially be targeted by these kinases (Bradshaw et al., 2013). Which isoform was tested here?

---

## [Author Response]

Essential revisions:1) Authors should look at other known NDEL1 interactions, and to determine whether the interactions described in this paper also occur with endogenous proteins.

In response to the reviewer’s suggestion regarding known NDEL1 interactors, we performed a series of co-immunoprecipitation (co-IP) experiments with LIS1 (*PAFAH1B1*) and DYNC1I1 (cytoplasmic dynein 1 intermediate chain 1). When we compared NDEL1 WT and NDEL1 S332/336A mutant, there was no significant difference in the interaction with LIS1 or DYNC1I1 (Figure 6—figure supplement 2, in the revised manuscript). We were not able to detect co-IP of TARA with either LIS1 or DYNC1I1 (Author response image 1). These results support the notion that the NDEL1 S336/S332 phosphorylation is less associated with microtubule dynamics.

**Author response image 1. respfig1:** coIP results to examine interactions between TARA and LIS1 or TARA and dynein intermediate chain.

To address the point regarding the protein interactions at the endogenous level, we conducted the co-IP experiments from HEK293 cell lysates and P7 mouse brain lysates. Endogenous TARA was detected in co-immunoprecipitates with endogenous GSK3β protein in the lysates of both developing brain and HEK293 cells (Author response image 2 and Figure 4E). DYRK2 was also co-IPed with TARA in HEK293 cell lysates (Figure 4D), but we could not test the IP in mouse brain lysates due to the species specificity of the antibody application. Overall, we believe that the interactions shown in overexpression conditions are mostly recapitulated at the endogenous protein level. With these additional data, we revised Figure 4.

**Author response image 2. respfig2:** The protein-protein interaction between endogenous GSK3β and endogenous TARA in the developing mouse brain.

2) Reviewers request genetic evidence, ideally knock in of the non-phosphorylatable or pseudo-phosphorylated residue to demonstrate in vivo evidence of the requirement of these events for neuronal morphogenesis. If not possible, at the minimum, the protein levels of NDEL1 or pNDEL1 (in the shRNA – overexpression rescue experiments) should be controlled to ensure physiological relevance.

In response to the reviewer’s comment, we designed the CRISPR-based knock-in construct targeting mouse *Ndel1* gene corresponding to exon 9 in order to substitute S332 and S336 to alanine residues and to fuse EGFP at the C-terminus of NDEL1 (Figure 2—figure supplement 2A). As a control, we also designed a donor DNA plasmid lacking S336/S332 mutations but still containing EGFP insertion. KI efficacy was validated in the NIH3T3 cell-lines (Figure 2—figure supplement 2B-C).

At first, we attempted to utilize these reagents for *in utero* electroporation at E15 embryonic brains. However, it was extremely challenging to collect the neurons in the electroporated brain that has proper EGFP expression, and thus we failed to obtain a meaningful number of neurons to analyze. We believe that the reasons are multiple layers; (1) All four constructs (dCas9^D10A^, two guide RNAs, and corresponding donor DNA plasmids) have to be successfully transfected. (2) The intrinsic knock-in rate is supposed to be also very low (Uemura et al., 2016; Yao et al., 2018). (3) The NDEL1 promotor appears to be very weak in that rare knock-in cells exhibited very weak fluorescence signals that were not readily distinguishable in the brain section. Combinations of these technical issues prohibited us from going further with this approach.

Alternatively, we applied these reagents to primary neurons. We co-transfected DIV 1 primary mouse hippocampal neurons with dCas9^D10A^, two guide RNAs, and corresponding donor DNA plasmids. When evaluated at DIV 4, axon/dendrite outgrowth was diminished in phospho-deficient (alanine mutant) knock-in neurons relative to wild-type (Figure 2G-I, Figure 2—figure supplement 1E-F). This result further strengthens our claim that NDEL1 S336/S332 phosphorylation impacts neuronal morphogenesis.

We also checked the potential issues related to the expression level of the transgene expressions. For this, the constructs expressing NDEL1 WT or S332/336A under UBC promoter, a low-expression promoter widely used in the fields, were generated. Western blot analysis confirmed that the UBC promoter drives considerably lower NDEL1 expression levels than CAG promoter (Figure 2—figure supplement 1G). The lower-expression of NDEL1^WT^ still effectively rescued NDEL1 knockdown effect in axon/dendrite outgrowth of primary rat hippocampal neurons, while NDEL1^S332/336A^ mutant failed to reverse the phenotype (Figure 2—figure supplement 1H-K). The low-expression NDEL1 constructs were also effective in the reversal of NDEL1 knockdown phenotype on dendritic arborization in P14 cortical layer II/III pyramidal neurons in the *in utero* electroporation (Figure 3—figure supplement 1L-N). Thus, we suppose that the expression level issue in our experimental setting does not seem to affect our interpretation of the results.

3) Is Ndel1 expressed and KD in neuronal (as opposed to glial) progenitors? To understand the function of pNDEL1 phosphorylation, it is necessary to understand cell type (progenitor/neuron/glia) and development stage-specificity of the S332/S336 phosphorylation. However, the phospho antibody also recognizes the native NDEL1, making it difficult to tease this out.

For axon/dendrite outgrowth assay from primary cultured hippocampal neurons, we analyzed the isolated cells with a significantly long single axon with multiple short and branched dendrites to exclude other cell types especially glial cells and undifferentiated cells. For dendritic arborization assay, cells located at the cortical layer II/III of somatosensory cortex were subjected to confocal imaging after determining this region based on Hoechst/DAPI nucleus staining patterns, the structure of hippocampus, and mouse brain atlas reference maps. Also, only cells with pyramidal cell-like soma shapes and proper apical dendritic structure (visible of stretching to the pial surface) were chosen for analyses. For this, we believe that the phenotype we see is likely a consequence of neuron-autonomous effects.

Attempting to address the reviewer’s point experimentally, we tried IUE in later stage, E17, where an *in utero* electroporation largely affects progenitors scheduled to be glial cells minimally affecting neuronal cell progenitors in the developing mouse brain (Kohwi and Doe, 2013; LoTurco et al., 2009; Miller and Gauthier, 2007; Taniguchi et al., 2012). When E17 embryos were subjected to IUE with NDEL1 shRNA constructs and analyzed at P14, affected cells were mostly localized around ventricular and subventricular zone (VZ/SVZ) and failed to reach to the cortical layers where most of our analyses focused on (Author response image 3). This confirms that the neuronal phenotypes characterized in this study were less likely to be attributable to glial cells.

**Author response image 3. respfig3:** Comparison of cell-types affected by *in utero* electroporation at either E15 or E17.

To examine the developmental stage-specificity of NDEL1 S336/S332 phosphorylation, we attempted to quantify pNDEL1 in the mouse brain lysates from multiple developmental stages. We enriched endogenous proteins by IP with pan-NDEL1 antibody for detection efficiency and measured the amount of S336/S332 phosphorylation by anti-pNDEL1 immunoblotting. In the result, the ratio of pNDEL1/NDEL1 peaked at in the developmental periods from E18 to P7 at which neuronal maturation and extensive axon/dendrite outgrowth and branching are to be active, supporting the importance of the phosphorylations in those process (Figure 1E in the revised manuscript).

4. The ideal experiment would be to demonstrate reduction of NDEL1 phosphorylation following kinase knockdown (in this paper, only kinase overexpression is performed). However, since this is probably quite challenging, the authors should at least (a) quantify the increase in pNDEL1 signal following kinase overexpression and (b) try to demonstrate the presence of this phosphorylation in another way (phos-Tag gel, MS?). It is not entirely clear to me why the FLAG-NDEL1 shows such a dramatic shift in the kinome experiment (sup Figure 1—figure supplement 1), but not endogenous NDEL1.

As the reviewer suggested, we quantified the increment of endogenous pNDEL1 signal after we replaced the Western blot images with improved quality (previously Figure 1G, now moved to Figure 1F). Since it was technically tricky to clearly distinguish endogenous pNDEL1 signal from the strong signal of antibody heavy chains, we utilized VeriBlot secondary antibodies (Abcam), which minimally bind to denatured IgG, and we were able to improve our pNDEL1 blot quality significantly. Indeed, we were able to detect a significant increment of endogenous NDEL1 S336/S332 phosphorylation in response to kinases over-expression.

Also, we elaborated the description of MS/MS analysis results of endogenous NDEL1 from mouse developing brain and over-expressed NDEL1 from HEK293 cells (Figure 1C-D, Figure 1—figure supplement 3, Figure 4B, and Figure 4—figure supplement 2).

To further solidify the results from Western blot and LC-MS/MS analysis, we examined the phosphorylation status of NDEL1 using phospho-serine residue-specific antibody (PhosphoSerine Antibody Q5, Qiagen). The NDEL1-specific phosphoserine signal was increased upon TARA over-expression (Figure 4—figure supplement 1C), supporting TARA-mediated phosphorylation of NDEL1.

The band shifts of NDEL1 by its phosphorylations have been reported by previous studies (Niethammer et al., 2000; Yan et al., 2003). To answer the reviewer’s comment regarding as “It is not entirely clear to me why the FLAG-NDEL1 shows such a dramatic shift in the kinome experiment (Figure 1—figure supplement 1), but not endogenous NDEL1”, we prepared an endogenous NDEL1 blot image with additional band shift by DYRK2-GSK3β over-expression (Author response image 4). In our experience, from 9~10% poly-acrylamide gel, FLAG-NDEL1 had considerable band shifts upon phosphorylations. Therefore, in order to detect the NDEL band shifts with high efficiency and sensitivity in the large scale kinome screening, we used FLAG-NDEL1 and FLAG antibody. Accordingly, we elaborated Materials and methods section.

**Author response image 4. respfig4:** ****Existence of endogenous NDEL1 band shift induced by DYRK2-GSK3β over-expression.

Additional references:

Kohwi M, & Doe CQ. (2013). Temporal fate specification and neural progenitor competence during development. Nat Rev Neurosci, 14(12), 823-838. https://www.ncbi.nlm.nih.gov/pubmed/24400340

Miller FD, & Gauthier AS. (2007). Timing is everything: making neurons versus glia in the developing cortex. Neuron, 54(3), 357-369. doi:10.1016/j.neuron.2007.04.019

Taniguchi Y, Young-Pearse T, Sawa A, & Kamiya A. (2012). In utero electroporation as a tool for genetic manipulation in vivo to study psychiatric disorders: from genes to circuits and behaviors. Neuroscientist, 18(2), 169-179. doi:10.1177/1073858411399925

Uemura T, Mori T, Kurihara T, Kawase S, Koike R, Satoga M, Cao X, Li X, Yanagawa T, Sakurai T, Shindo T, & Tabuchi K. (2016). Fluorescent protein tagging of endogenous protein in brain neurons using CRISPR/Cas9-mediated knock-in and in utero electroporation techniques. Sci Rep, 6, 35861. doi:10.1038/srep35861

Yan X, Li F, Liang Y, Shen Y, Zhao X, Huang Q, & Zhu X. (2003). Human Nudel and NudE as regulators of cytoplasmic dynein in poleward protein transport along the mitotic spindle. Mol Cell Biol, 23(4), 1239-1250. doi:10.1128/mcb.23.4.1239-1250.2003

Yao X, Zhang M, Wang X, Ying W, Hu X, Dai P, Meng F, Shi L, Sun Y, Yao N, Zhong W, Li Y, Wu K, Li W, Chen ZJ, & Yang H. (2018). Tild-CRISPR Allows for Efficient and Precise Gene Knockin in Mouse and Human Cells. Dev Cell, 45(4), 526-536 e525.

doi:10.1016/j.devcel.2018.04.021

[Editors' note: further revisions were requested prior to acceptance, as described below.]

The manuscript has been improved but there are some remaining issues that need to be addressed before acceptance, as outlined below. The authors provided thorough responses to the critiques and added high quality new data to address many questions raised by reviewers. The new actin-based mechanism shown by this study appears independent of NDEL1's brain partners, and the developmental analyses of the revised manuscript remain partially unclear due to the lack of specificity of the experiments. The remaining issues could be addressed experimentally or through better description of the possible interpretation or limitations of the approaches applied.1) Authors acknowledged that the NDEL1 promoter is very weak. In this case, it is concerning to use even the UBC promoter, as there was no information to justify the level of expression delivered by electroporation. Moreover, the effect of loss of NDEL1/pNDEL1 function on neuronal morphology was assessed by knockdown electroporation at E15, but it is not clear how effect knockdown or replacement worked, or the cell types affected (i.e. NPC, glial progenitor, etc).

We apologize for this confusion from our response to the essential comment #2. We wanted to mean by the statement that EGFP-NDEL1 expression under endogenous NDEL1 promoter was not strong enough to be used for the analysis of dendritic morphology of KI neurons in the brain slice culture, which does not necessarily mean that the endogenous expression of NDEL1 is too weak to be functional in the brain development. Indeed, in cultured primary neurons on coverslips, we were able to analyze the neuronal morphology after KI (Figure 2G-I). Also, when we carry out immunohistochemistry with anti-pNDEL1 antibody, we found endogenous pNDEL1 signal was diminished by NDEL1 knockdown in the electroporated neurons (Figure 1—figure supplement 2D). Moreover, quantitative analysis of NDEL1 S336/S332 phosphorylation in mouse brain lysates from various developmental stages indicated the significant level of NDEL1 phosphorylation encompassing E15, E18, and P7 (Figure 1E). We believe that these observations can support the impact of knockdown and replacement constructs on dendritic arborization after that electroporation at E15.

We agree with the reviewer’s concern that NDEL1 expression by UBC promoter is still overexpression condition, although it is significantly lower than CAG promoter. We would like to cautiously emphasize that overexpression by CAG promoter and UBC promoter induced indistinguishable strength of phenotype. Moreover, the overexpressed NDEL1^S332/336A^ under CAG or UBC promoters still failed to rescue NDEL1 knockdown effect on the neuronal morphology. Furthermore, when we over-express NDEL1 alone by CAG promoter, there was no significant effect on neuronal morphogenesis despite its high expression level (Author response image 5), which further supports the notion that the roles for NDEL1 phosphorylations in the neuronal morphogenesis shown in this work are less likely due to an overexpression artifact. Accordingly, we revised our manuscript to reflect this notion (see subsection “Phosphorylation of NDEL1 S336/S332 regulates neuronal morphogenesis”).

**Author response image 5. respfig5:** Neuronal morphogenesis with single over-expression of TARA, NDEL1^WT^, or NDEL1^S332/336A^.

Existence of endogenous NDEL1 band shift induced by DYRK2-GSK3β over-expression

While we think that the results including cultured primary neurons suggest that our observations are relatively neuron-autonomous effect, we also agree that we cannot fully exclude the contribution of neuron-nonautonomous effects in the *in utero* electroporation approach. Accordingly, we revised our manuscript to reflect this notion (see subsection “Phosphorylation of NDEL1 S336/S332 regulates neuronal morphogenesis”).

2) Reviewers remain unconvinced that there is no impact on MTs. The microtubule co-sedimentation assay (Author response image 6) suggests that in the presence of overexpressed GFP-TARA, there is more flag-pNDEL1in the soluble fraction than in the absence overexpressed GFP-TARA. This suggests that the interaction between TARA and NDEL1 might impact the interaction of NDEL1 and MTs.

We agree with the reviewer's concerns in that the basis of our claim that the NDEL1 S336/S332 phosphorylation effect is rather specific to the actin-related events is the limited aspects of microtubule functionalities, which does not allow us to exclude the possibility of NDEL1 S336/S332 phosphorylation-dependent microtubule functions. In regards to Author response image 6, the result could be interpreted in multiple ways and cannot be the basis of excluding the additional roles of the phosphorylation in NDEL1-MT interaction. Accordingly, we revised our manuscript toning down the claim related to this point wherever necessary (Abstract, subsection “Phosphorylation of NDEL1 S336/S332 enhances F-actin dynamics”, and the Discussion section).

**Author response image 6. respfig6:** 

3) The lysosome trafficking study did not include NDEL1 alone. It is possible that TARA might impact the effect of NDEL1 with respect to organelle trafficking. For example, NDEL1 alone might increase retrograde organelles more than NDEL1S332/336A regardless of TARA, or both WT and mutant NDEL1 could impact transport more in the absence or TARA. This is relevant to the interesting model shown in Figure 7 and may suggest that TARA and the kinases regulate an interplay between F-actin- and MT- associated NDEL1.

While we were preparing the first revision regarding Figure 6—figure supplement 3, we initially tested the NDEL1 alone and NDEL1^S332/336A^ alone in a preliminary set of experiments and we were not able to see the difference between WT and mutant NDEL1 group in the fraction of motile lysosome. We thought the capacities of WT and mutant NDEL1 to impact on lysosome trafficking were not different in this experimental setting potentially due to stoichiometric imbalance between NDEL1 and TARA, so we switched to TARA co-expression condition to sensitize the potential impact of the phosphorylation as in other assays we used (Figure 5). Indeed, the lysosome phenotype remained indistinguishable in this condition. Again, we agree that we cannot exclude the possibility that TARA-mediated phosphorylation modulates the association of NDEL1 with microtubule. Accordingly, we further elaborated the description of related part in the manuscript. (Discussion section).

4) Please explain the mass spec data from brain (Figure 1D) and HEK293 cells overexpressing NDEL1 and TARA (Figure 4B). From these figures it seems to indicate that S332 is phosphorylated in brain but not in the HEK293 cells. Is that correct?

We apologize for the less clear presentation of the data. To clarify, we were able to recover an ionized fragment that predicts pS336 from the HEK293 sample, and we recovered an ionized fragment corresponding to pS332 or pS336 from the mouse brain sample. Due to technical limitations of LC-MS/MS technique and S/T-rich nature of NDEL1 carboxyl terminus, we had to admit that MS data is not conclusive enough to definitely claim the dual phosphorylation of S332 and S336 on its own. However, in conjunction with other biochemical data including phospho-specific antibody data and mutagenesis data, etc., this data supports our finding that S332 and S336 are targets of phosphorylation.

For better description of the data, we elaborated on the LC-MS/MS data in Figure 1D and Figure 4B and corresponding descriptions (subsection “DYRK2 and GSK3β induce sequential phosphorylation of NDEL1 at S336 and S332”, and subsection “TARA recruits DYRK2 and GSK3β to induce sequential phosphorylation of NDEL1 S336/S332”).

5) The SSSSC splice variant of NDE1 could potentially be targeted by these kinases (Bradshaw et al., 2013). Which isoform was tested here?

Here, we used a mouse NDE1 with canonical sequence (UniProtKB/Swiss-Prot ID: Q9CZA6-1, CCDS ID: CCDS37263.2). This isoform has similar size (human/mouse NDEL1: 345 a.a., mouse NDE1: 344 a.a.) but different C-terminal sequences with human/mouse NDEL1 protein (Please see Figure 1 of Bradshaw et al., 2013). Accordingly, we added this information in subsection “Antibodies and plasmids”.